# Liquid metals for boosting stability of zeolite catalysts in the conversion of methanol to hydrocarbons

Yong Zhou [1,2], Sara Santos[3], Mariya Shamzhy [4], Maya Marinova[5], Anne-Marie Blanchenet[6], Yury G. Kolyagin [1], Pardis Simon [5], Martine Trentesaux[1], Sharmin Sharna[7], Ovidiu Ersen [7], Vladimir L. Zholobenko[8], Mark Saeys [3], Andrei Y. Khodakov [1] ✉ & Vitaly V. Ordomsky [1] ✉

Methanol-to-hydrocarbons (MTH) process has been considered one of the most practical approaches for producing value-added products from methanol. However, the commonly used zeolite catalysts suffer from rapid deactivation due to coke deposition and require regular regeneration treatments. We demonstrate that low-melting-point metals, such as Ga, can effectively promote more stable methanol conversion in the MTH process by slowing coke deposition and facilitating the desorption of carbonaceous species from the zeolite. The ZSM-5 zeolite physically mixed with liquid gallium exhibited an enhanced lifetime in the MTH reaction, which increased by a factor of up to ~14 as compared to the parent ZSM-5. These results suggest an alternative route to the design and preparation of deactivation-resistant zeolite catalysts.

Nowadays, the catalytic conversion of source-abundant methanol molecules via methanol-to-hydrocarbons (MTH) reaction is considered one of the most promising approaches for producing valuable chemicals and fuels[1–4]. Among them, light olefins (ethylene, propylene, and butylene) and aromatics are highly desirable since they could act as essential industrial feedstocks for synthesizing polymers, pharmaceuticals and chemicals, etc[4–6]. According to the latest results, the MTH reaction proceeds according to a double-cycle mechanism with the production of paraffins and olefins by the olefin cycle and ethylene and aromatics by the aromatics cycle[7]. The specific products of MTH can be tuned into olefin-rich (methanol-to-olefins; MTO) or aromatic-rich (methanol-to-aromatics; MTA) mixtures by optimizing catalysts and reaction conditions[1]. Zeolites, such as ZSM-5 and SAPO-34, are commonly used as MTH catalysts[8]. However, they suffer from rapid deactivation because of the

formation of bulk coke compounds and require regular regeneration treatments by calcination[9–11].

In order to obtain long-term stable methanol conversion as well as a specific product range during MTH, many strategies have been developed, such as optimizing the reaction conditions by using a high-pressure process with co-feeding hydrogen and water[12]; tuning zeolite chemical composition[13,14] as well as acid sites location/and distribution[15]; reducing zeolite crystal dimensions by utilizing layered[16] or nanosized zeolites[17]; introducing secondary porosity via mesoporous engineering[18], etc. In addition, metal-modified zeolites, for instance, Zn- and Ga-modified ZSM-5, have proven to offer high selectivity to aromatic products and are also expected to improve the catalytic stability during the MTH process[19–22]. Advanced NMR and FTIR characterizations revealed that both cationic metal species (e.g., $Ga^{3+}$, $GaO^+$, $Ga^+$) and metal oxides (e.g., $Ga_2O_3$) act as Lewis acid sites

[1]Univ. Lille, CNRS, Centrale Lille, ENSCL, Univ. Artois, UMR 8181 – UCCS – Unité de Catalyse et Chimie du Solide, F-59000 Lille, France. [2]Research Institute of Interdisciplinary Sciences (RISE) and School of Materials Science & Engineering, Dongguan University of Technology, Dongguan 523808, China. [3]Laboratory for Chemical Technology (LCT), Department of Materials, Textiles and Chemical Engineering, Ghent University, Technologiepark 125, 9052 Ghent, Belgium. [4]Department of Physical and Macromolecular Chemistry, Faculty of Science, Charles University, Hlavova 2030/8, 12843 Prague, Czech Republic. [5]Institut Michel-Eugène Chevreul, 59655 Villeneuve-d'Ascq, France. [6]Univ. Lille, CNRS, UMR 8207—UMET—Unité Matériaux et Transformations, Lille F-59000, France. [7]IPCMS, Université de Strasbourg—CNRS, 67034 Strasbourg, France. [8]School of Physical and Chemical Sciences, Keele University, Staffordshire, UK. ✉e-mail: andrei.khodakov@univ-lille.fr; vitaly.ordomsky@univ-lille.fr

and facilitate aromatization reactions over Ga-modified zeolites[23–25]. Framework Ga in zeolite also proved favorable to aromatization reactions[26]. However, high selectivity towards aromatic intermediates usually leads to more polycyclic aromatic products, which may cause catalyst deactivation. To address this disadvantage of the Ga-modified ZSM-5 catalyst, Kapteijn and co-workers modified it with calcium, which was proven to stabilize the intrazeolite gallium oxide clusters and moderate their dehydrogenation activity. As a result, the bimetal (Ca, Ga) modified ZSM-5 exhibited improved stability and showed a high yield of aromatic in the MTH process[19]. Notwithstanding the progress in the improvement of Ga-zeolite activity, selectivity and stability in methanol conversion, to the best of our knowledge, only the effects of framework Ga ions and supported $GaO_x$ have been investigated.

Metals or alloys with low melting points (e.g., Ga, Bi, In, etc.)[27] have been widely used as dopants for supported metal to form intermetallic catalysts[28,29] or as electronic and structural promoters[30–32]. In the latter references, it has been demonstrated that the appropriate surrounding of low-melting-point metal not only maximizes dispersibility but also tunes the electronic structure of the active metal sites, therefore enhancing the catalytic activity and long-term stability of the catalyst in various applications, such as low-pressure Fischer-Tropsch synthesis[33], alcohol amination[34], etc. Recently, Wasserscheid et al. proposed a fascinating concept, the supported catalytically active liquid metal solutions (SCALMS)[35]. In a series of pioneering works, they demonstrated that supported Ga-rich M-Ga phases (M = Pd, Pt, Rh) present as a liquid under reaction conditions (typically in the temperature range of 350–500 °C) can offer excellent catalytic activity and coke resistance in propane and butane dehydrogenation[35–37]. Rahim et al. developed the liquid Pt catalyst using a solvent-like Ga matrix at low temperatures. They showed that an ultralow content of Pt (~0.0001 at%) dissolved in Ga could elevate the mass activity to at least three orders of magnitude higher than any existing Pt catalyst for methanol oxidation at 70 °C[38].

Although considerable research is related to the application of liquid metal for modification of the metal catalysts, to the best of our knowledge, there have been no reports focused on the modification of non-metallic catalysts using liquid metals. In this work, the catalytic performance of ZSM-5 catalysts promoted by liquid metals has been explored in the MTH process. We have found that the presence of liquid Ga physically mixed with ZSM-5 can significantly improve the stability of catalyst with an increase in its lifetime from 8 to 110 h under the same conditions. Detailed characterization suggests that liquid Ga decorates zeolite crystals modifying and facilitating the desorption of coke species from the acid sites (Fig. 1).

## Results and discussion
### Enhancing MTH catalytic stability by liquid metal
The catalytic performances of the parent ZSM-5 and Ga with ZSM-5 zeolites in the MTH reaction are illustrated in Fig. 2. The reference ZSM-5 and $Ga_2O_3$/ZSM-5 catalysts have shown a similar lifetime (defined as a period during which the methanol conversion remains stable at nearly 100%) of 8–9 h (Fig. 2a). In contrast, both the Ga+ZSM-5 prepared by physically mixing ZSM-5 with metallic gallium and Ga/ZSM-5 prepared by heat treatment of ZSM-5 with gallium have demonstrated a substantially improved catalytic stability with a lifetime of ~32 h.

At the initial period after about 8 h of testing, the main products formed over ZSM-5 were light paraffins (27%), $C_5$+ hydrocarbons (20%) and aromatics (42%) with a small contribution from light olefins (7%). $Ga_2O_3$/ZSM-5 shows a higher selectivity to aromatics and a lower selectivity towards $C_5$+ hydrocarbons in agreement with an earlier publication[3] demonstrating a promoting effect of Ga oxide towards aromatization reactions. In contrast, the presence of liquid Ga in the reactor for both Ga+ZSM-5 and Ga/ZSM-5 increases the catalyst selectivity to light olefins and aromatics with a decrease in the formation of $C_5$+ hydrocarbons as compared with pure ZSM-5. The catalytic performance changes over time with an increase in the production of olefins and a decrease in the production of light paraffins and aromatics due to a decrease in the hydrogen transfer activity of the catalysts (Fig. S1, SI). It has to be noted that Ga in the mixture with inert silica does not provide catalytic activity in methanol conversion.

The catalytic performance of ZSM-5-based catalysts in MTH with hydrogen co-feeding is also presented in Fig. 2. ZSM-5 and $Ga_2O_3$/ZSM-5 have shown insignificant effects of hydrogen on the catalyst lifetime, which is due to the use of atmospheric pressure. Indeed, high hydrogen pressure is required for the hydrogenation of carbon species inducing the deactivation of the catalyst[12]. In contrast, Ga+ZSM-5 and Ga/ZSM-5 have demonstrated significantly improved catalytic stability with little deactivation up to ~70 and 110 h of time-on-stream, respectively. In addition, compared to ZSM-5, the selectivity to aromatics is higher over Ga+ZSM-5 and Ga/ZSM-5, while a larger amount of paraffin is formed due to the hydrogenation reactions.

The effect of other liquid metals and alloys on the catalytic performance of ZSM-5 has been tested in the presence of indium (In), bismuth-indium (BiIn) and bismuth-tin (BiSn) alloys with melting points at 156, 62, and 138 °C, respectively (Fig. S2, SI). There is almost no effect of In on the catalyst lifetimes. BiIn and BiSn have increased the MTH stability from 8 h to 19 and 20 h, respectively. The presence of hydrogen did not affect significantly the catalytic stability of BiSn +ZSM-5 in comparison with Ga+ZSM-5 (Fig. S2, SI).

The zeolite changes color from white to gray after treatment with liquid metal such as Ga with small droplets of Ga not absorbed by zeolite (Fig. S3, SI). Ga is changing catalyst color by interaction with zeolite, however, it is still in the form of metal covered by a thin oxide layer. The presence of Ga in the metal state can be confirmed by TG analysis, where it shows an endothermic peak during melting at 33 °C (Fig. S4, SI). In the case of zeolite mixed with Ga and deposited in the reactor, metallic Ga on the wall of the reactor can be observed after the reaction (Fig. S5, SI). The chemical analysis of Ga/ZSM-5 and Ga+ZSM-5 separated from Ga droplets by sieving shows that they contain 24 and 14.9 wt. % Ga, respectively. The higher metal loading and more uniform distribution of Ga in Ga/ZSM-5 could explain the higher stability of the material in the reaction in comparison with Ga+ZSM-5. It should be noted that Ga+ZSM-5 exhibited the highest Ga content after reaction and sieving off bulk gallium droplets, surpassing other metal- and alloy-promoted zeolites (with In at 0.1 wt.%, BiSn at 0.9 wt.%, and BiIn at 5.1 wt.%). This is in line with the assumption about the stronger effect of Ga on the catalytic performance in the MTH reaction (Fig. S6, SI).

The regeneration of deactivated zeolite by calcination in air is traditionally used in the MTH process. In the case of Ga/ZSM-5 catalyst, the calcination in the air could oxidize Ga to $Ga_2O_3$ with a loss of enhanced catalytic stability. It requires additional reduction treatment to reduce oxidized gallium back to the metallic state. Our results show that the deactivated Ga/ZSM-5 catalysts can be regenerated in 3 reaction cycles without losing the catalytic performance (Fig. S7, SI).

### Role of liquid Ga in improving the zeolite stability
To understand the promotion effect of liquid Ga for ZSM-5 in the MTH, ZSM-5, Ga+ZSM-5, and Ga/ZSM-5 catalysts have been characterized using a broad range of techniques. According to X-ray diffraction (XRD) patterns, the MFI zeolite phase was observed in both ZSM-5 and Ga/ZSM-5 before and after the catalytic tests (Fig. S8, SI), suggesting that Ga did not modify the zeolite framework. STEM-HAADF and elemental mapping images of as-prepared Ga/ZSM-5 show the decoration of zeolite crystals by small-size Ga nanoparticles (Fig. 3a−c). Cutting of the sample by cryo-ultramicrotome demonstrates penetration of Ga to the distance of only about 20 nm in zeolite crystal (Figs. 3d−f, S9, SI), which could be explained by diffusion limitations for deeper

penetration of Ga inside of the pores. It can be observed that other liquid metals, e.g., BiSn alloy, have lower penetration ability in comparison with Ga (Fig. S10, SI). Ga metal is highly mobile in the reactor, which results in modifying the zeolite surface and subsurface layer and thus, affecting its MTH performances. The lowest atomic radius of Ga (1.35 Å) in comparison with other liquid metals (In: 1.93 Å, Sn: 2.17 Å, Bi: 1.63 Å) and low viscosity of Ga (Ga: 1.016 mPa s, In: 1.748 mPa s) at 177 °C[39] can provide the highest access to the micropores of ZSM-5 (5.4 Å × 5.6 Å).

We also used in situ TEM to trace the Ga species morphology in the Ga/ZSM-5 sample (Fig. S11, SI). During the activation with $N_2$ and exposure to the methanol at the reaction temperature, the initial Ga droplets are deformed and redispersed to form smaller Ga particles or Ga-containing species on the surface and at the entrance of zeolite pores.

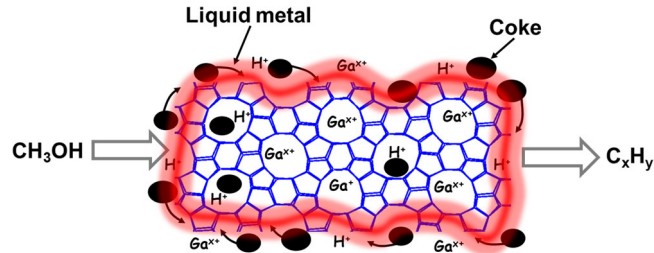

**Fig. 1 | Scheme of the effect of liquid metal on the MTH reaction by desorption of carbon species.**

Density Functional Theory (DFT) calculations were performed to develop a description on a molecular level of the stability of small Ga clusters inside the ZSM-5 pores, and the effect of acid sites. The stability of Ga clusters with an increase of the size from 1 to 4 atoms increases, converging to the sublimation energy for bulk gallium, $-290$ kJ/mol$_{Ga}$ (Table S1, SI). The interaction of the small Ga clusters ($Ga_1$–$Ga_4$) with ZSM-5 and with Silicalite-1 (pure Si form, no Al sites) was computed (Fig. 4, Figs. S12–S14, SI). Small Ga clusters interact quite strongly and specifically with the acid protons in the zeolite (Table S1, SI), with an adsorption energy of $-116$ kJ/mol$_{cluster}$ for $Ga_1$ and $-143$ kJ/mol$_{cluster}$ for $Ga_4$. The introduction of Ga also influenced the location of the acid proton. Instead of an H-bond with a neighboring O-atom, the proton now points towards the large pore to interact with Ga (Fig. 4). The adsorption energy is much weaker in the Silicalite-1, e.g., $-58$ kJ/mol$_{cluster}$ for $Ga_1$ (Table S1, SI). This specific interaction of Ga atoms and small clusters with acid sites of the zeolite can explain the partial penetration of Ga in the pores of zeolite.

The deactivation of zeolite catalyst in MTH is mainly ascribed to the carbonaceous deposition which deactivates acid sites and blocks the entrance of micropores. TG analysis (Figs. 2c, S15, SI) performed on spent ZSM-5, Ga/ZSM-5 and Ga+ZSM-5 after 20 h on stream shows two stages of the weight loss (10 wt% in total) in ZSM-5_20h at about 100 and 550 °C, which can be ascribed according to the literature[40] to the desorption of water and burning of condensed the graphitic coke species, respectively. Ga/ZSM-5_20h and Ga+ZSM-5_20h also show the formation of graphitic coke, however, the weight loss is about ~4 times lower than that for ZSM-5_20h. This suggests that Ga can suppress the formation of coke species over ZSM-5 and slow catalyst deactivation.

This conclusion has been further supported by $N_2$ adsorption on ZSM-5 and Ga+ZSM-5 (Fig. 2d). The ZSM-5 after $N_2$ activation at 450 °C

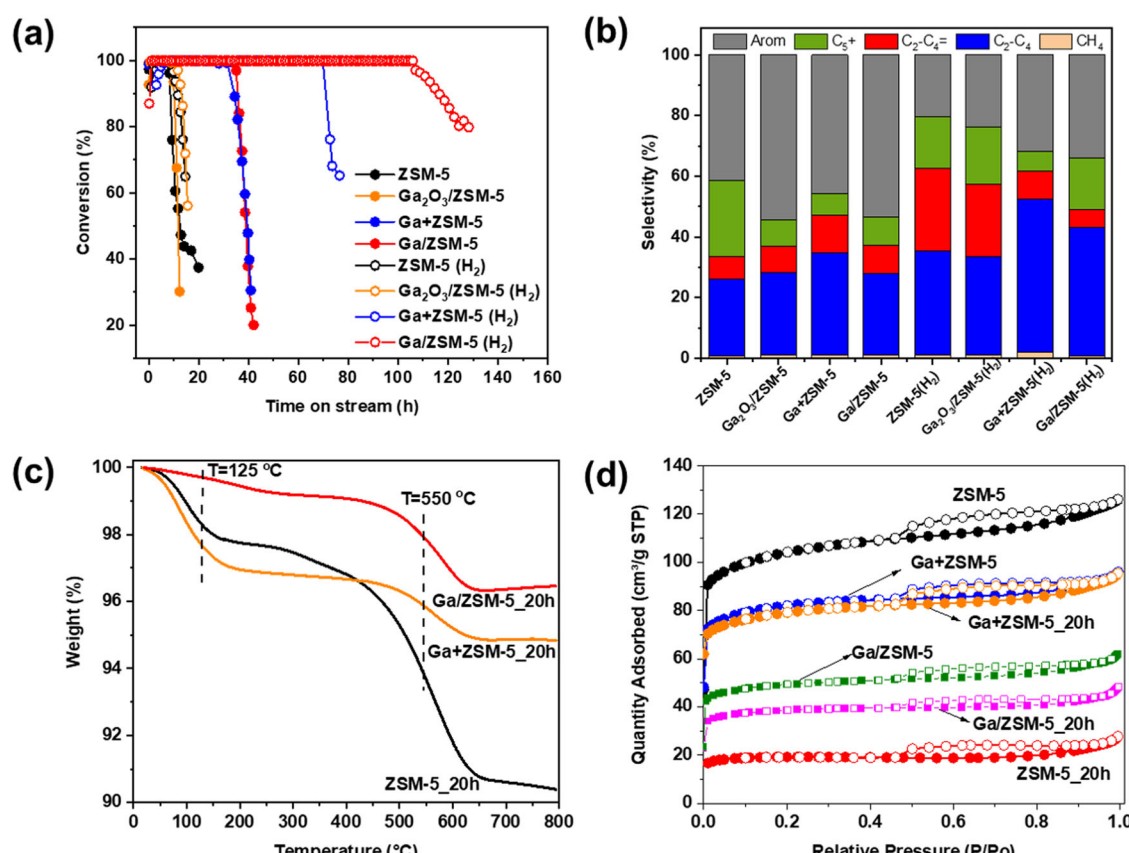

**Fig. 2 | Effect of liquid Ga promotion of ZSM-5 on MTH reaction.** Catalytic conversion (**a**) and selectivity to the products after 8 h of reaction (**b**) of ZSM-5, $Ga_2O_3$/ZSM-5, Ga+ZSM-5 and Ga/ZSM-5 in the methanol-to-hydrocarbon reaction. Reaction conditions: 400 °C, catalyst containing 50 mg of ZSM-5, 1.4 g methanol g$^{-1}_{ZSM-5}$ h$^{-1}$. TGA analysis (**c**) and low-temperature $N_2$ adsorption (**d**) of initial and after 20 h on stream ZSM-5, Ga+ZSM-5, and Ga/ZSM-5.

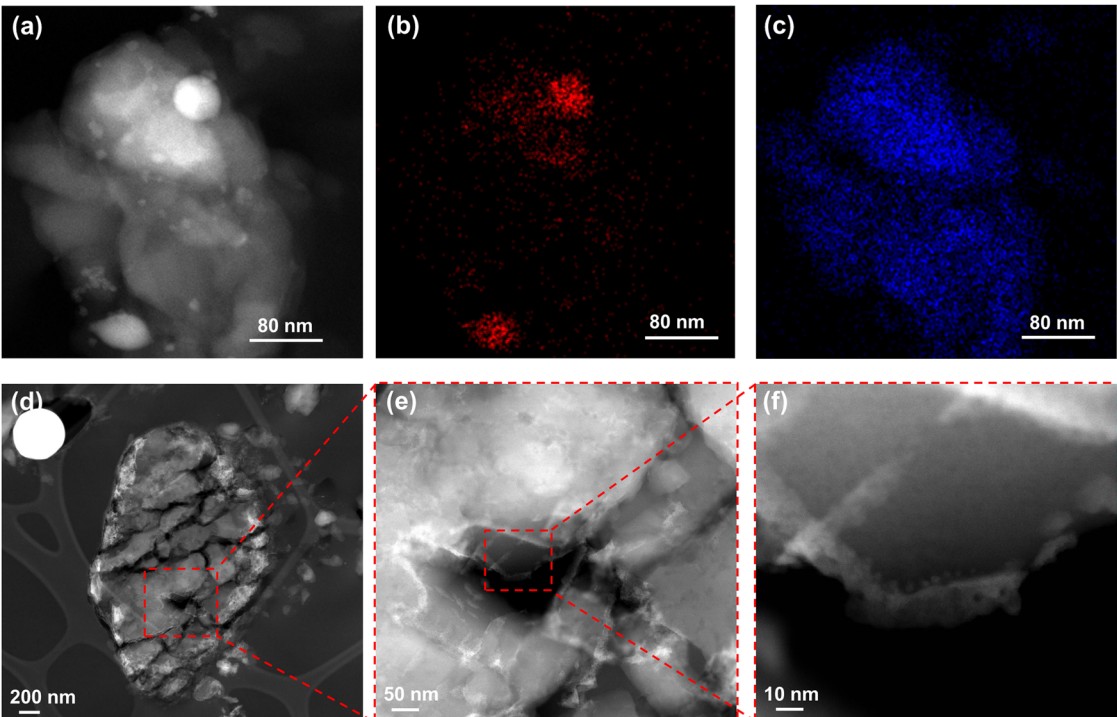

**Fig. 3 | Penetration of Ga in ZSM-5 studied by STEM-HAADF.** STEM-HAADF image (**a**) with EDS elemental mapping of Ga (**b**) and Si (**c**) and STEM-HAADF images after cutting with ultramicrotome at liquid nitrogen temperature (**d**–**f**) of Ga/ZSM-5 sample.

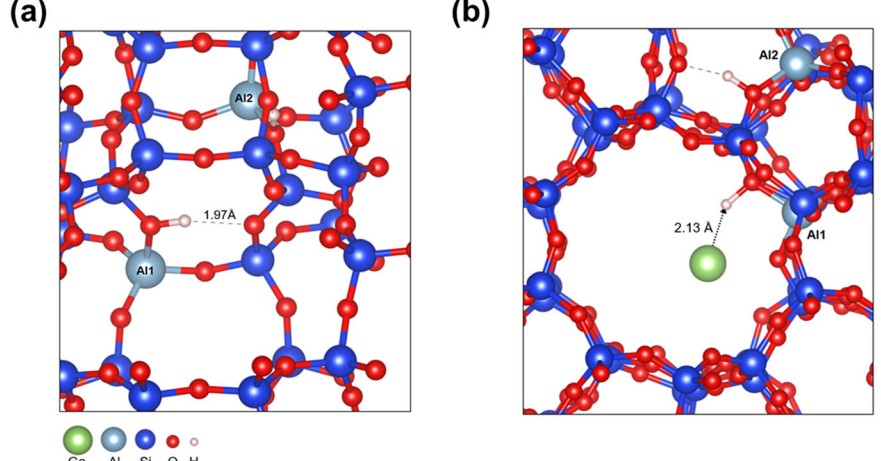

**Fig. 4 | The simulated interaction between Ga with acid sites on zeolites.** Structure of acid sites (**a**) in ZSM-5 and Ga$_1$/ZSM-5 (Si/Al = 47, 2 Al sites) showing the interaction of Ga with the acid site (**b**).

for 1 h possesses a surface area of 339 m²/g, while the Ga+ZSM-5 and Ga/ZSM-5 samples show a decrease in the surface area to 266 and 160 m²/g, respectively, due to the introduction of Ga species into the pores of zeolite and dilution effect of zeolite with metallic gallium. Coke deposition over ZSM-5 after 20 h resulted in a significant decrease in the surface area to 62 m²/g. In contrast, the surface area is almost retained on Ga+ZSM-5_20h and Ga/ZSM-5_20h.

Given that the addition of Ga improves the catalytic stability and prevents the coke deposition of ZSM-5 catalysts in MTH, it is hypothesized that Ga could be also used to regenerate the coked ZSM-5 catalyst after methanol conversion. To verify this assumption, we have collected the deactivated ZSM-5 after a 20-h reaction, mixed it with Ga and tested it in the MTH reaction (Fig. S16, SI). It is observed that after the treatment with Ga, the deactivated ZSM-5 recovered its activity and

showed ~100% methanol conversion for 4 h. Thus, we may assume that mobile Ga can remove coke species and liberate acid sites in MTH.

The additional visual evidence that liquid metal could suppress carbon deposition was supported by HAADF imaging and elementary analysis for both ZSM-5 and Ga/ZSM-5 used after 20 h. Figures 5 and S17, SI show a uniform distribution of carbon species in the crystal of ZSM-5_20h zeolite. It is interesting to note that in the presence of Ga, carbon is mainly localized at the intercrystalline zone mixed with Ga nanoparticles (Figs. 5b,e, S18, SI). Microscopy analysis of deactivated ZSM-5 after Ga treatment shows a significant change in the distribution of carbon in comparison with the parent ZSM-5_20h sample (Figs. 5c,f, S19, SI). There is more carbon localized in the intercrystalline zone between zeolite crystals together with Ga. It is interesting to note that large Ga particles present in the sample after

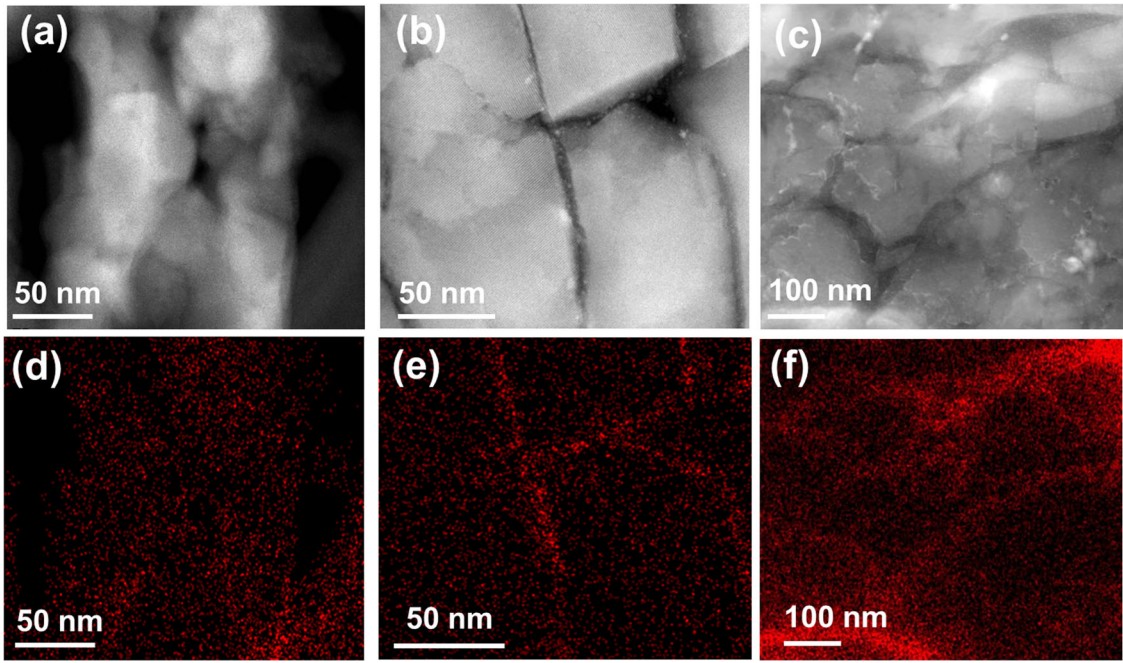

**Fig. 5 | Coke localization studied by EDS elemental mapping.** Representative HAADF (**a**–**c**) and EDS elemental mapping of C (**d**–**f**) of ZSM-5_20h (**a,d**), Ga/ZSM-5_20h (**b, e**) and ZSM-5_20h+Ga (**c, f**).

treatment also contain carbon on the surface (Fig. S20, SI) indicating the high affinity of carbon to metallic Ga.

The localization of carbon in the intercrystalline zone with Ga after the reaction could be due to the extraction of coke from the zeolite, but this hypothesis is less likely due to the limited penetration capacity of Ga into the zeolite pores. Another explanation could be a stable catalytic performance mainly in this intercrystalline zone, which explains the coke generation. The role of Ga could involve continuous regeneration of acid sites in this area through the elimination of coke species until Ga and acid sites are completely saturated by carbon.

This hypothesis that Ga prevents the coke deposition by promoting the desorption of aromatics has been further checked by in situ diffuse reflectance infrared Fourier transform (DRIFT) spectra for ZSM-5 and Ga/ZSM-5 samples. Variable temperature IR spectroscopy has been carried out in the temperature range of 30–225 °C by using in situ IR cell (Fig. S21, SI). Adsorption of methanol on the catalyst surface with subsequent heating results in a gradual increase of the peak intensity at 1453 cm$^{-1}$ assigned earlier to aromatic C=C stretching vibrations (Fig. S22, SI)[41,42]. It should be noted that Ga/ZSM-5 catalyst does not show the formation of this peak.

In addition, the pre-adsorption of toluene as a model aromatic molecule over ZSM-5 and Ga/ZSM-5 shows peaks at 2800–3100 cm$^{-1}$ related to C-H stretching and 1300–1500 cm$^{-1}$ related to C–C stretching in the aromatic ring and asymmetric and symmetric bending vibrations of the methyl group (Fig. 6a). The amount of toluene adsorbed is significantly lower over Ga/ZSM-5 in comparison with ZSM-5 with fast desorption of toluene in the flow of Ar in comparison with less significant desorption over ZSM-5 (Fig. 6b). These results demonstrate that the presence of Ga suppresses the adsorption of the aromatic products and enhances their desorption once they are formed in the zeolite catalyst in the MTH reaction, thus enhancing its stability.

The effects of gallium on the acid sites in ZSM-5 and Ga/ZSM-5, their concentration and strength, have been determined by pyridine-FTIR (Fig. 6c, d and Table S2, SI). The parent ZSM-5 contains intense IR adsorption bands of Brønsted acid groups (3606 cm$^{-1}$) associated with the framework Si-OH-Al bridges, isolated external silanols (3745 cm$^{-1}$) and Al-OH groups (3678 cm$^{-1}$)[43]. The intensity of all OH bands

decreases considerably when the ZSM-5 is modified with Ga (Ga/ZSM-5). The Py adsorption on ZSM-5 results in the appearance of peaks corresponding to Brønsted (BAS, 1545 cm$^{-1}$) and Lewis acid sites (LAS, 1456 cm$^{-1}$) with concentrations of 0.82 and 0.22 mmol/g, respectively (Fig. 6d). By introducing Ga, the concentration of BAS is almost halved without a significant effect on the amount of LAS. The amount of acid sites decreases almost by a factor of 4 over ZSM-5 after 20 h of the reaction, whereas Ga/ZSM-5_20h has only 1.6 times less BAS and the same number of LAS when compared with the initial Ga/ZSM-5 catalyst.

The decrease in the amount of Brønsted acid sites after the introduction of Ga could be attributed to the oxidation of Ga to cationic Ga species (Ga$^+$, GaO$^+$) which should be accompanied by the generation of hydrogen. Treatment of ZSM-5 by Ga in the batch reactor at 250 °C under a 10 bar N$_2$ atmosphere shows the formation of 1.6 mmol/g of hydrogen, which could be formed by partial oxidation of Ga with acid sites of zeolite and water in zeolite (Figs. S23 and S24, SI). Usually, the introduction of Ga by conventional impregnation results in the appearance of Lewis acid sites due to the presence of cationic Ga. The absence of new Lewis acid sites (Fig. 6d) could be explained by the interaction of formed Lewis sites with atoms of Ga in the pores resulting in their deactivation and possibly plugging of zeolite pores by Ga clusters. Indeed, according to DFT modeling penetration of metallic Ga inside of the pores could be in the form of small clusters.

Figure S25, SI shows the Ga 3$d$ X-ray Photoelectron Spectroscopy (XPS) spectra of the Ga catalysts before and after the reaction. Compared to Ga$_2$O$_3$/ZSM-5 containing only Ga$_2$O$_3$, both gallium oxide and metallic gallium are present in Ga+ZSM-5 after physical mixing due to the oxidized surface of metallic Ga. The amount of oxidized Ga phase increases after the reaction for both Ga+ZSM-5 and Ga/ZSM-5, which could be assigned to the oxidation of Ga phase by generated water. Hydrolysis of gallium cations or oxidation of metallic Ga on the surface could result in the formation of gallium oxide. The partial oxidation of Ga can be also observed by analysis of the heat flow during TG analysis (Fig. S4, SI). The negative peak at 33 °C corresponds to the melting of Ga in the sample, which is accompanied by an exothermic peak at 470 °C due to the oxidation of Ga to oxide. The reaction in the N$_2$

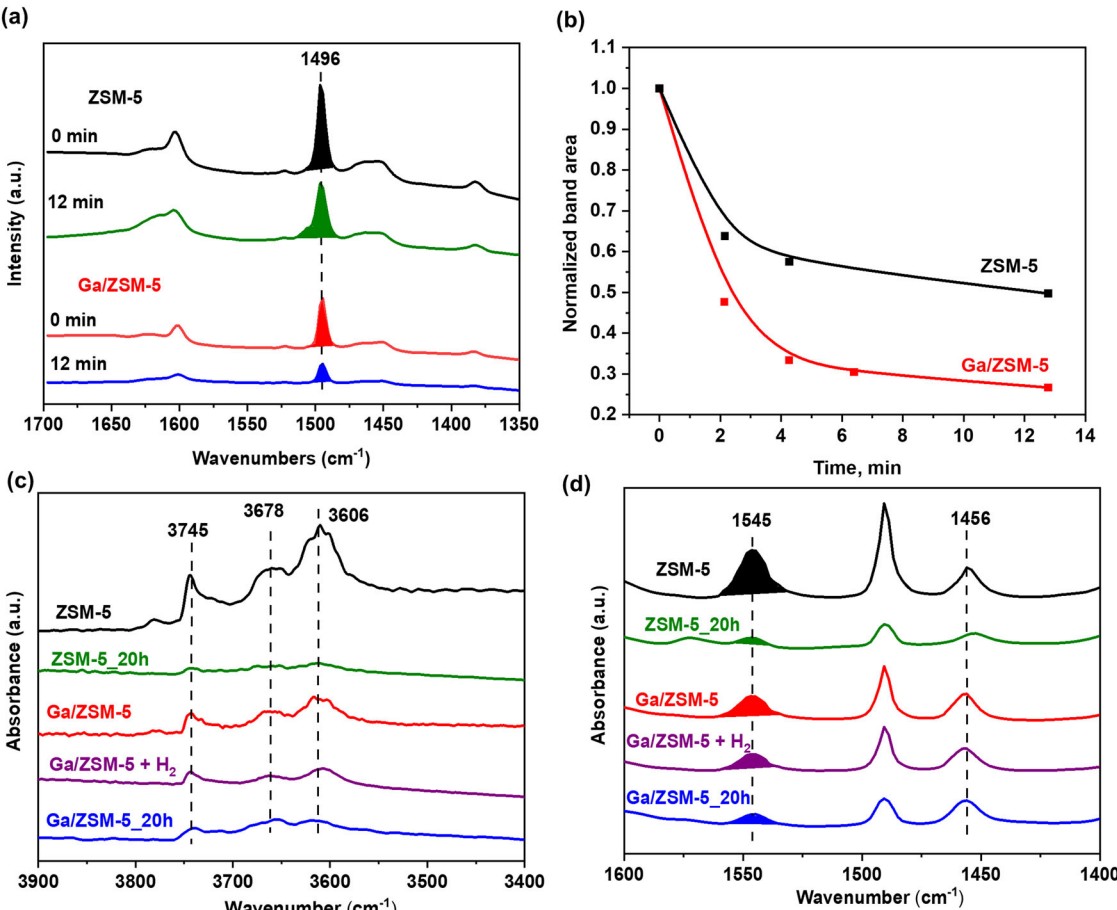

**Fig. 6 | In situ FTIR measurements of ZSM-5 and Ga/ZSM-5.** In situ toluene desorption DRIFT spectra over ZSM-5 and Ga/ZSM-5 catalysts at room temperature for 0 and 12 min (**a**). The residual percentage of toluene according to the peak-intensity changes (signal at ~1496 cm⁻¹) over ZSM-5 and Ga/ZSM-5 catalysts for 0–12 min (**b**). FTIR spectra of activated and spent ZSM-5 and Ga/ZSM-5 catalysts with and without H$_2$ treatment: region of hydroxyl groups vibrations (**c**) and region of pyridine adsorption vibrations (**d**).

atmosphere makes these peaks less noticeable, however, they are even more obvious in the presence of H$_2$ indicating the reduction of Ga. These results, however, support our assumption that in addition to carbon deposition, the deactivation of Ga-modified ZSM-5 could be also induced by the oxidation of Ga by water during the catalytic reaction. It should suppress the regeneration effect of metallic Ga on the acid sites. This also suggests that the MTH stability potentially could be further enhanced by co-feeding methanol with hydrogen. H$_2$-TPR of Ga$_2$O$_3$/ZSM-5 shows reduction starting only at about 400 °C (Fig. S26, SI). However, Ga/ZSM-5 shows reduction already at ~360 °C corresponding to smaller gallium oxide particles or gallium oxide over the surface of metallic Ga with a broad peak till 650 °C. Note that the catalytic stability in the presence of hydrogen has been significantly improved for both Ga/ZSM-5 and Ga+ZSM-5 (Fig. 2). XPS analysis confirms a significant increase in the contribution of metallic Ga after the reaction in the presence of hydrogen in comparison with the pure oxide phase over Ga$_2$O$_3$/ZSM-5, which supports our hypothesis about the importance of metallic Ga for stable catalytic performance (Fig. S25, SI). It is interesting to note that cationic Ga is most probably not affected by hydrogen due to the same amount of adsorbed Py after pretreatment of the Ga/ZSM-5 catalyst in H$_2$ at 450 °C (Fig. 6c,d).

Based on these results, we could assume that oxidized Ga should not contribute significantly to the increase in the stability of ZSM-5 during MTH reaction. However, these species affect the selectivity of the reaction in N$_2$ atmosphere, resulting in an increased yield of aromatics for Ga$_2$O$_3$/ZSM-5 and Ga/ZSM-5 compared to ZSM-5, which

could be the result of the dehydrogenation activity on Ga oxide (Fig. 2). Ga+ZSM-5 provides lower selectivity to aromatics most probably due to less uniform distribution of Ga in the sample and lower content of oxidized Ga in this case. Interestingly, the hydrogenation activity on Ga/ZSM-5 and Ga+ZSM-5 is considerably enhanced in the presence of hydrogen with higher paraffins production compared to ZSM-5 and Ga$_2$O$_3$/ZSM-5 (Fig. 2). This could indicate hydrogenation behavior of cationic Ga in ZSM-5 or oxidized Ga on the Ga metal surface.

According to TEM and FTIR results, metallic Ga localized over the external surface of zeolite and at the entrance of the pores effectively promotes more stable methanol conversion in MTH process by slowing deposition and facilitating the desorption of carbon species. The internal acid sites should contribute less to the catalytic performance due to blockage of the pores by liquid Ga and their deactivation by reaction with Ga. At the same time, Ga at the external surface of zeolite provides continuous refreshing of the acid sites in the intercrystallite voids by removing coke species and liberating the acid sites. This effect could be attributed to the high affinity of metallic Ga to carbon species (Fig. 5), which has been observed in the literature for different applications[44,45]. It should result in the pushing out carbon species and the release of acid sites of ZSM-5. The interaction of Ga with acid sites of ZSM-5 according to DFT modeling should also promote the desorption of carbon species (Fig. 4). It is also supported by FTIR spectroscopy indicating a decrease in the strength of interaction between acid and adsorbed organic compounds in the presence of Ga. The dynamic nature of Ga promotes the migration of carbon at the

interface of zeolite crystals, which can be observed by microscopy. This method of stability enhancement in the presence of liquid metal may act as an alternative route to design more stable high-temperature processes for zeolite catalysts.

We have demonstrated boosted stability of the MTH process on a ZSM-5 catalyst conducted in the presence of liquid metal. Multiple characterizations revealed that liquid Ga is localized in the inter-crystalline area and partially penetrates in the ZSM-5 micropores, modifies the zeolite acidic properties, slows down carbon deposition and facilitates the desorption of aromatic molecules. Moreover, it has been found that co-feeding with hydrogen could enhance the MTH lifetime by a factor of ~14 compared to ZSM-5. We believe that the facile mixing of the zeolite catalyst with low-melting-point metal can act as a powerful promoter for the MTH process, which could also open an alternative route to the design and preparation of a more efficient zeolite catalyst system for other high-temperature reactions.

## Methods

### Materials

$NH_4$-ZSM-5 (CBV2314) with a $SiO_2/Al_2O_3$ ratio of 23 was provided by Zeolyst. Methanol (anhydrous, 99.8%), Ga (99.99% trace metals basis), In (99.99% trace metals basis) and $Ga(NO_3)_3$ (gallium(III) nitrate hydrate, 99.9% trace metals basis) were purchased from Sigma-Aldrich. BiSn (Bi 58%-Sn 42%, melting point at 138 °C) and BiIn alloys (Bi 67%-In 33%, melting point at 109 °C) were purchased from Haines & Maassen Metallhandelsgesellschaft mbH.

ZSM-5 catalysts for the MTH reaction were prepared via calcination of $NH_4$-ZSM-5 at 550 °C (ramp of 2 °C/min) for 6 h in static air. $Ga_2O_3$/ZSM-5 with a nominal 10%wt Ga loading was prepared via incipient wetness impregnation with an aqueous solution of $Ga(NO_3)_3$. After the impregnation, the sample was first dried at 80 °C overnight and then calcined at 550 °C (ramp of 2 °C/min) under static air for 6 h. X-ray fluorescence (XRF) analysis showed ~6.6%wt of Ga in the $Ga_2O_3$/ZSM-5 sample. Ga/ZSM-5 was obtained by mixing the 50 mg of ZSM-5 with metal Ga in a weight ratio of 1:1 then stirring and heating the mixture at 250 °C under 10 bar of $N_2$ overnight. The heating was used to facilitate gallium migration over the zeolite. After the treatment, the unsupported Ga droplets were separated by sieving using a 250 μm stainless steel mesh sieve, the rest of the powder sample was collected and denoted as Ga/ZSM-5. Ga+ZSM-5 was produced by physically mixing 125 mg of metal Ga with 50 mg of ZSM-5 with no further treatment before catalytic usage. Other catalysts with liquid metals such as In, BiIn and BiSn have been prepared in the same way as with Ga using the same mass of metal.

### Catalyst characterization

XRD data were recorded using a PANalytical Empyrean X-ray diffractometer with CuKα radiation (40 kV and 30 mA) used as the X-ray source.

XRF analysis of the chemical composition was performed using an energy-dispersive micro-X Ray Fluorescence spectrometer M4 TORNADO (Bruker). This instrument is equipped with two anodes: a rhodium X-ray tube (50 kV, 600 mA, 30 W) and a tungsten X-Ray tube (50 kV, 700 mA, 35 W). For sample characterization, the rhodium X-ray tube with a polycapillary lens enabling excitation of an area of 200 μm was used. The silicon-drift-detector Si(Li) with a resolution of 145 eV at 100,000 cps (Mn Kα) with a Peltier cooling (−20 °C) was utilized. The measurements were carried out under a vacuum.

$N_2$ adsorption isotherms were collected using a volumetric gas adsorption analyzer (Quantachrome Instruments Autosorb-iQ-MP-AG) at −196 °C. Typically, 50–80 mg powder sample was loaded in a 6 mm large bulb sample cell and degassed under vacuum at 120 °C for 8 h. The BET surface area was determined using the data points in the pressure range of 0.01–0.1$P/P_0$. The spent catalysts were analyzed by

thermogravimetric analysis on a Q600 SDT instrument. In a typical measurement, a small amount of a spent catalyst was heated in an $Al_2O_3$ crucible from ambient temperature to 800 °C at 10 °C/min in a stream of air at a flow rate of 100 mL/min.

The hydrogen temperature programmed reduction ($H_2$-TPR) of the catalysts was evaluated using an AutoChem II 2920 apparatus (Micromeritics). The samples (~0.05 g) were reduced in a flow of 5% $H_2$/Ar (50 mL/min) and heated to 900 °C with a temperature ramp of 10 °C/min.

The scanning transmission electron microscopy (STEM) was performed on a TITAN Themis 300 S/TEM microscope equipped with a probe aberration corrector, allowing a spatial resolution of 70 pm, a super-X windowless 4 quadrant SDD (silicon drift detector) detection system for STEM-EDX mapping and several annual dark field detectors. Measurements were performed with a spot size of about 500 pm, a semi-convergence angle between 20 mrad, and a probe current of ~100 pA. For HAADF images, collection angles were chosen between 50 and 200 mrad. The EDX mapping was performed in spectrum imaging mode, with dwell time per pixel of about 10 μm in continuous scanning of multiple frames over 10–15 min. The samples were prepared by ultramicrotomy by embedding the powders in epoxy for the sample before reaction. In order to exclude carbon-containing epoxy the samples after the reaction were prepared by diluting them in water and then freezing them with liquid nitrogen, followed by cutting with cryo-ultramicrotome.

In situ STEM experiments were carried out using a JEOL JEM-2100F microscope at 200 kV equipped with a spherical aberration probe corrector, high-resolution objective lens pole piece and an UltraScan 1000 CCD array detector (from GATAN). For the in situ observation of the Ga migration, the Ga/ZSM-5 sample was used. The in situ STEM experiments were carried out using an enclosed micro-electro-mechanical systems system from Protochips[46]. The setup includes a sample holder capable of heating and introducing a gas environment, a gas manifold to introduce the desired gas and a vapor kit to produce methanol vapor. The in-situ observations were carried out at a pressure of 1 bar either under pure $N_2$ or with a mixture of 11% methanol vapor in $N_2$ at a flow rate of 0.1 mL/min. The sample was initially heated from 25 °C to 450 °C at a rate of 20 °C/min, under $N_2$ and kept for 2 h. Next, the temperature was decreased to 400 °C and a reaction mixture of methanol/$N_2$ was introduced for 4 h. Several areas were selected to be observed at regular time intervals, typically every 30 min. Additional observations were made to determine the maximum electron beam dose and to detect any sample damage or beam-induced artefacts.

FTIR transmission measurements of pyridine, methanol and toluene adsorption on zeolite catalysts were performed using a Thermo iS 50 spectrometer equipped with a DTGS detector. Zeolites were pressed into self-supporting wafers with a density of ~10 mg/cm² and activated in an in situ transmission cell at $T = 450$ °C in vacuum for 4 h. The additional reduction has been performed if necessary at 450 °C and 100 Torr of $H_2$. Pyridine adsorption was carried out at 150 °C for 20 min at a partial pressure of 3 Torr, followed by desorption for 20 min at the same temperature or at 350 °C. Toluene adsorption was carried out at 30 °C for 20 min at a partial pressure of 1 Torr, followed by desorption at the same temperature monitored by FTIR spectroscopy over time. Methanol was dosed into the cell to reach the partial pressure of 1 Torr; the cell was closed and transmission FTIR spectra were recorded while the temperature was increased at 3 °C/min. All spectra were recorded with a resolution of 4 cm⁻¹ by collecting 89 scans for a single spectrum at room temperature (for pyridine and toluene) or at 30–225 °C (for methanol). Spectra were re-scaled using a wafer density of 10 mg/cm².

Concentrations of Lewis ($C_L$) and Bronsted ($C_B$) acid sites were evaluated based on the FTIR spectra of adsorbed pyridine from the integral intensities of the bands at 1456 cm⁻¹ ($C_L$) and at 1545 cm⁻¹ ($C_B$)

using molar absorption coefficients, $\varepsilon(L) = 1.71\ cm\ \mu mol^{-1}$, and $\varepsilon(B) = 1.098\ cm\ \mu mol^{-1}$[47].

The XPS experiments were carried out using a Kratos Axis Ultra DLD spectrometer, equipped with a monochromatic Al Kα X-ray source (1486.6 eV) operating at 225 W (15 kV, 15 mA). The instrument base pressure was $5 \cdot 10^{-10}$ torr and the charge neutralizing system was used for all acquisitions. Analysis was performed at a pass energy of 40 eV and a step size of 0.1 eV. Binding energies (BE) were referenced to Si 2p peak (unresolved doublet) positioned at 103.5 eV.

## Catalytic test

Prior to the catalytic tests, the catalysts were loaded into a fixed bed quartz tube reactor with a diameter of 6 mm and activated at 450 °C under flowing $N_2$ (20 mL/min) for 40 min, after which the temperature was set to 400 °C for the reaction. Methanol was fed into the reactor at atmospheric pressure by passing a carrier gas ($N_2$, 20 mL/min) through a saturator containing methanol at 25 °C. In the case of the test with $H_2$, 18 mL/min of $N_2$ was mixed with 2 mL/min of $H_2$ before entering the reactor with a ratio of $H_2$ to methanol of about 2. All gas lines after the reactor were heated to 100 °C, and the reaction product analysis was carried out using a Bruker GC456 GC equipped with a Rt-Q-BOND column (30 m, 0.32 mm ID, 10 μm) and an FID. The methanol conversion and the selectivities for various products were calculated on carbon basis. Both methanol and dimethyl ether were considered reactants owing to their rapid interconversion[40]. In the case of regeneration treatment, the catalyst has been calcined in the flow of air at 500 °C during 6 h with subsequent reduction by hydrogen at the same conditions.

## DFT modeling

Periodic DFT calculations were carried out using the Vienna Ab Initio Simulation Package[48–50]. Geometry optimization of the structures was performed using the Perdew–Burke–Ernzerhof functional with Grimme D3 dispersion corrections and a Becke-Johnson damping function (BJ)[51,52]. A projector-augmented wave method and a plane wave basis set with a kinetic cutoff energy of 600 eV were applied[53,54]. The self-consistent field convergence criterion was set at $10^{-5}$ eV. The ZSM-5 and Silicalite-1 frameworks and the Ga clusters were fully relaxed during geometry optimizations.

## Data availability

The data of this study are available from the corresponding author upon request.

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

## Acknowledgements

M.S. acknowledges the Czech Science Foundation for project 20-12099S. S. Santos and M. Saeys acknowledge the financial support for a doctoral fellowship grant (1SA4124N) from the Flemish Research Foundation (FWO), and the UGent High-Performance Computing (HPC). The computational resources (Stevin Supercomputer Infrastructure) and services used in this work were provided by the VSC (Flemish Supercomputer Center), funded by Ghent University, FWO and the Flemish Government—department EWI. The authors thank the Chevreul Institute (FR 2638) for its help in the development of this work. Chevreul Institute and the Microscopy Platform in Lille is supported by the Ministre de l'Enseignement Superieur et de la Recherche et de l'Innovation, the CNRS, the Region Hauts-de-France, the Metropole Europeenne de Lille, and the Fonds Europeen de Developpement des Regions. V.O. acknowledges the financial support of the French National Research Agency (DEZECO, Ref. ANR-22-CE05-0005).

## Author contributions

V.O. and A.K. conceived the idea for the research. Y.Z., A.K. and V.O. wrote the manuscript. Y.Z., S.Sharma, M.Shamzhy, M.M., P.S. carried out the experimental measurements. M.Saeys and S.Santos built and analyzed the theoretical model and performed the quantum calculations. Y.Z., S.Santos, M.Shamzhy, M.M., A.B., Y.K., P.S., M.T., S.Sharma, O.E., V.Z., M.Saeys, A.K. and V.O. discussed, validated and analyzed the data and improved the manuscript. All authors reviewed and contributed to the final manuscript.

## Competing interests

The authors declare no competing interests.
