## [Peer Review File · Nature Communications]

REVIEWER COMMENTS

Reviewer #1 (Remarks to the Author):

The manuscript "Liquid Metals for Boosting Stability of Zeolite Catalysts in the Conversion of Methanol to Hydrocarbons" by Zhou et al. describes the use of Ga-modified ZSM-5 catalysts for the MTH reaction. Rather than introducing the Ga by framework incorporation or impregnation/ion exchange of salts, the authors make use of the low melting point of Ga to introduce the metal as a liquid in the system. The approach is interesting and novel to my knowledge. I have two major points of criticism: the first is that I see no convincing evidence the Ga is actually in the pores of the zeolite (2D TEM is not sufficient to prove incorporation, and the XRD suggests there is nothing in the pores), and second : it is imperative for MTH operation that the catalyst can be regenerated and re-used in multiple catalytic cycles. I would expect an oxidate regeneration to be detrimental to the catalyst. more detailed comments follow below.

Line 26-27: "(catalysts...) require regular regeneration treatments". I see no regeneration results that could help establish whether the catalysts can be regenerated.

Line 57: The authors seem to suggest Ga in general introduces Lewis Acid sites. The present system, however, does not (as the authors clearly show).

Line 68: "it is demonstrated that the appropriate surrounding of low melting-point metal not only maximizes dispersibility but also tunes the electronic structure of the active metal sites": I would very much like to see some description on a molecular level inside the zeolite pores of what the authors think is the active site in this system, and what is the role of the Gallium.

Line 80: "the application of liquid metal": I know the system is operating above the melting point of Gallium, but what proof is there that the Ga is actually in liquid state?

Line 89 Figure 1. I believe the cartoon-description of the coke species inside the zeolite pores is misleading. The smallest aromatic molecule (i.e. coke molecule) is benzene, which is about 5 Å in size, which is almost the entire radius of the zeolite pore. Coke precursors in MTH are believed to get stuck inside the zeolite pores and thereby block the pore system. That does not change here. Additional coke deposits form outside the crystal and block the pore entrances. I do believe the authors demonstrate this latter coke may be reduced, but I do not believe the coke-precursor transport by the liquid Ga.

Line 104: "The unsupported Ga particles were removed by sieving" could you provide some more detail?

Line 124: "the BET surface area was determined": Is BET a good method to probe the pore volume of microporous zeolites?

Line 147: reference 35. Something went wrong, there is no information in the reference list.

Line 184: "In the case of the test with H₂, 18 mL/min of N₂ was mixed with 2 mL/min of H₂." : When exactly was this mixture used? Also during heating? Can the authors provide an estimate of the H₂/MeOH ratio?

Line 200, figure 2A. There are interesting differences in the induction periods. Can you elaborate a bit more on that?

Line 202: "50 mg of catalyst or 50 mg of ZSM-5 mixed with 125 mg of liquid metal" This is not very clear. Does that imply there is always 50 gram of zeolite? Or should we read this another way?

Line 227: Why were the other metals not tested with H₂ in the feed?

Line 236: "It should be noted that Ga+ZSM-5 exhibited the highest Ga content following the reaction". Does this imply some of the liquid metals disappear? Why, where? Boiling points are much higher than the process temperatures.

Line 246: "penetration of Ga into the zeolite crystal": I do not believe the 2D TEM is convincing proof of penetration into the pores. The XRD peaks at low angle are almost unchanged, which would indicate there is no incorporation of any significant species into the pores.

Line 249: " the lower atomic radius" surely the atomic radius is not the only parameter that influences whether an atom in a liquid will enter pores. What about surface energy, contact angle, viscosity, etc.?

Line 255: "undetectable at the TEM resolution". Earlier on the page you indicate some of the Ga goes missing, is that not a more logical explanation for not seeing it in TEM?

Line 262: why was the Ga/ZSM-5 not used?

Line 272: Same, why was the Ga/ZSM-5 not used for this comparison?

Line 280: This is not the experiment I would have used to make any statements about regeneration. Adding Ga may help remove the coke outside the crystals, but not the coke inside the crystals.

Line 290: "slowing down coke deposition in the micropores". That would be enough by itself, the "washing out" is not necessary to explain the better performance (and in my opinion, not a logical explanation).

Line 332. The formation of hydrogen indicates removal of BAS sites. Does the addition of hydrogen during the experiment reinstate them?

Line 343 The TPR-temperature are measured in a dynamic system, and can not be directly compared to the reaction conditions.

Line 356: "It should result in the pushing out of carbon species and the release of acid sites of ZSM-5." I do not believe this. Ga outside the crystal may help removing carbon extra-crystalline carbon species. The only results on acid sites you show indicate a drop in acid sites.

Line 359: "The dynamic nature of Ga promotes further migration of carbon from the inside of the pores to the external surface of zeolite crystals" This would imply there is a driving force for Ga to move to the outside of the crystal.

Line 367: I do not believe you have provided convincing evidence that the Ga is inside the zeolite pores.

Line 269: This is a very nice result, but we would need to see the effect of oxidative regeneration.

Line 370: "facial" do you mean "facile"?

Line 463: Information reference 35 is missing

Supporting information

In general: supporting text explaining what it is we see or are supposed to be seeing would be helpful.

Figure S1: consider using the same x-axis for all plots. Why does ZSM-5H₂ shows such a long induction period? Where are the results for Ga/ZSM-5 and Ga+ZSM-5 with H₂?

Figure S4: I believe you mention you have indications for Ga-deposits on the reactor walls. Where do we see that?

Figure S5: Please explain the observations in the SI text. Where does all the other metal go?

Figure S6. The ratio of intensities for the peaks below 10 degrees 2 theta to the other peaks does not seem to change. That is a clear indication the Ga is not in the pores.

Figure S8. Please explain what we are supposed to look for

Figure S9. I see no difference between these three images. 3D TEM or tilt series would be more convincing.

Figure S10. Why is Ga/ZSM-5 not shown?

Figure S11: Is the amount zeolite in both experiments the same? Or is the zeolite diluted in the Ga+used ZSM-5 case?

Figure S16: I would consider showing only the relevant parts of the GC-spectra. The blue text (?) is unreadable, and may be irrelevant. Consider re-designing this graph.

Figure S17: Why does it take so long before we see hydrogen. Why do we see hydrogen with neat ZSM-5? Can it be something else? Do you have GCMS to prove it is H₂? or just GC retention times?

Figure S18: (e): why does the metallic Ga go down with H₂-addition? (g): where is the second O 2s peak?

In conclusion, the work is novel and relevant, but does require quite some revision before it could be considered for publication.

Reviewer #2 (Remarks to the Author):

The manuscript titled "Liquid Metals for Boosting Stability of Zeolite Catalysts in the Conversion of Methanol to Hydrocarbons" addresses an intriguing topic in catalysis. The authors explore the use of liquid metals to enhance the stability of zeolite catalysts during the conversion of methanol to hydrocarbons, liquid metal seems to have become the armor of ZSM-5 zeolites, preventing surface coking, and even has a certain reduction effect for the already coked zeolites, such effect was assigned to the high affinity of Ga to carbon, which is quite compelling. This study presents a novel approach by incorporating liquid metals into the catalytic system, and this concept has the potential to significantly improve the stability of zeolite catalysts, which is a critical challenge currently. The use of liquid metals as promoters offers new avenues for enhancing the catalytic performance and extending the catalyst lifetime. This innovation opens up possibilities for developing more efficient and robust catalytic systems for methanol conversion. However, there are still some queries which should be clarified:

1. In page 6 line 103-109, the only difference between the preparation of Ga/ZSM-5 and Ga+ZSM-5 appears to be that Ga+ZSM-5 is prepared without heating and sieving procedure. Could you explain the purpose of heating? Some research had reported that metal oxides seem soluble in gallium under specific condition. Does external heat promote the fusion of ZSM-5(SiO₂ or Al₂O₃) with gallium?
2. Fig S3 shows that the Ga/ZSM-5 catalyst completely lost the metallic luster of gallium, but the powder looked moist and varied in particle size. I wonder if there is a grinding or sieving process before filling it in the quartz tube.
3. In page 18 line 334-343, what is the proximate factor in the deactivation of ZSM-5 due to gallium oxidation?

Response to reviewers

We thank the reviewer for the comments to this manuscript. We made point-by-point response to these comments and revised the manuscript accordingly. All major changes have been highlighted in the manuscript and SI.

Reviewer #1

The manuscript "Liquid Metals for Boosting Stability of Zeolite Catalysts in the Conversion of Methanol to Hydrocarbons" by Zhou et al. describes the use of Ga-modified ZSM-5 catalysts for the MTH reaction. Rather than introducing the Ga by framework incorporation or impregnation/ion exchange of salts, the authors make use of the low melting point of Ga to introduce the metal as a liquid in the system. The approach is interesting and novel to my knowledge. I have two major points of criticism: the first is that I see no convincing evidence the Ga is actually in the pores of the zeolite (2D TEM is not sufficient to prove incorporation, and the XRD suggests there is nothing in the pores), and second: it is imperative for MTH operation that the catalyst can be regenerated and re-used in multiple catalytic cycles. I would expect an oxidate regeneration to be detrimental to the catalyst. more detailed comments follow below.

Reply: Thank you for the positive evaluation of our work. We have tried to reply to these two main points of the reviewer, which can be found later in the response letter.

Comment 1. Line 26-27: "(catalysts...) require regular regeneration treatments". I see no regeneration results that could help establish whether the catalysts can be regenerated.

Reply 1: The catalyst can be regenerated by calcination in the air with subsequent reduction in the hydrogen. The three consecutive catalytic tests demonstrate the high stability of the catalyst.

Revision 1 (P12): *"The regeneration of zeolite after deactivation by calcination in air is traditionally used in the MTH process. In the case of Ga/ZSM-5 catalyst, the calcination in the air could oxidize Ga to Ga₂O₃ with the loss of enhanced catalytic stability. It requires additional reduction treatment after calcination to reduce gallium*

back to the metallic state. It provides comparable stability for three cycles (Figure S7, SI).”

Figure S7. The catalytic tests over Ga+ZSM-5 for 3 cycles with intermediate calcination at 500 °C for 6h and reduction in hydrogen flow at the same conditions. Reaction conditions: 400 °C, catalyst containing 50 mg of ZSM-5, 1.4 g methanol · g⁻¹_{ZSM-5} · hour⁻¹.

Comment 2. Line 57: The authors seem to suggest Ga in general introduces Lewis Acid sites. The present system, however, does not (as the authors clearly show).

Reply 2: Indeed, the substitution of protons by Ga should increase the contribution of Lewis acid sites. However, according to the literature this effect is observed during impregnation of ZSM-5 with Ga salts (Ga(NO₃)₃) and subsequent calcination. The introduction of Ga in the form of metal could result in the penetration of clusters of Ga according to DFT modelling which should result in deactivation of the formed Lewis Ga⁺ or Ga³⁺ by another atom of Ga.

Revision 1 (P20): “Usually, the introduction of Ga by conventional impregnation results in the appearance of Lewis acid sites due to the presence of cationic Ga. The absence of new Lewis acid sites (Figure 6d) could be explained by the interaction of formed Lewis sites with atoms of Ga in the pores resulting in their deactivation and

possibly plugging of zeolite pores by Ga clusters. Indeed, according to DFT modelling penetration of metallic Ga inside of the pores should be in the form of clusters.”

Comment 3. Line 68: "it is demonstrated that the appropriate surrounding of low melting-point metal not only maximizes dispersibility but also tunes the electronic structure of the active metal sites": I would very much like to see some description on a molecular level inside the zeolite pores of what the authors think is the active site in this system, and what is the role of the Gallium.

Reply 3: DFT modelling has been used to study the interaction of acid sites in ZSM-5 with Ga. To our surprise, Ga clusters interact quite strongly and specifically with the acid protons in the zeolite. It could be the reason for the strong effect of Ga on the catalytic performance of zeolite explaining its enhanced stability.

Revision 3: (P14) *“DFT calculations were performed to develop a description on a molecular level of the stability of small Ga clusters inside the ZSM-5 pores, and the effect of acid sites. The stability of Ga clusters with an increase of the size from 1 to 4 atoms increases, converging to the sublimation energy for bulk Gallium, -290 kJ/mol_{Ga} (Table S1, SI). The interaction of the small Ga clusters (Ga₁ to Ga₄) with ZSM-5 and with a silicalite (pure Si form of ZSM-5, no Al sites) was computed (Figure 4, Figure S12-S14, SI). Small Ga clusters interact quite strongly and specifically with the acid protons in the zeolite (Table S2, SI), with an adsorption energy of -116 kJ/mol for Ga₁ and -143 kJ/mol for Ga₄. A similar strong interaction was not observed for Ga with HCl (-41 kJ/mol), so the origin of this strong interaction remains to be elucidated. The introduction of Ga also influenced the location of the acid proton. Instead of an H-bond with a neighbouring O-atom, the proton now points towards the large pore to interact with Ga (Figure 4). The adsorption energy is much weaker in the silicalite, e.g., -58 kJ/mol for Ga₁ (Table S2, SI). This specific interaction of Ga atoms and small clusters with acid sites of the zeolite can explain the partial penetration of Ga in the pores of zeolite.”*

Figure 4. Structure of acid sites (a) in H-ZSM-5 and Ga₁/ZSM-5 (Si/Al=47, 2 Al sites) showing the interaction of Ga with the acid site (b).

Figure S12. Periodic H-ZSM-5 model with 2 Al sites with corresponding protons (Si/Al=47), front and side view. In the most stable structure, the protons form a H-bond interaction with neighboring O-atoms.

DFT calculations were performed using a periodic H-ZSM-5 model with a Si/Al ratio of 47 and a unit cell of 96 T atoms. The Brønsted acid sites are introduced by substituting Si by an Al atom and a charge compensating proton on a neighboring oxygen. The acid sites are in an 8-membered ring at the intersection of the straight and sinusoidal channels (**Figure S12**). These positions have been identified as favorable positions for Al substitution.¹

Figure S13. Reactions used to compute the adsorption energy of Ga_n cluster on HZSM-5 pores (**R1**) and on pure-silica ZSM-5 (no Al sites) (**R2**).

The stability of small metallic Ga clusters (Ga_n) relative to bulk α -Gallium was calculated using PBE-D3(BJ) DFT electronic energies, as implemented in VASP.^{2,3} Ga_1 to Ga_4 clusters in their most stable configuration were computed.

Figure S14. Structure of $Ga_1/HZSM-5$ ($Si/Al=47$). The interaction of Ga with the acid proton is shown.

Introduction of small Ga clusters in pure silica ZSM-5 hence has a high energy penalty of 232 kJ/mol_{Ga} for Ga_1 and 111 kJ/mol_{Ga} for Ga_4 . This energy penalty can be compensated by entropy-gain at high temperatures, TDS, depending on the size of the cluster. The strong interaction between Ga clusters and the acid sites reduces this energy penalty for HZSM-5 to 174 kJ/mol for Ga_1 and 101 kJ/mol for Ga_4 and hence reduces the temperature for which the energy penalty is compensated by entropy gain, TDS. More detailed calculations are required to compute this entropy gain, in particular for small Ga_2 and Ga_3 clusters.

Table S1. The formation energy of Ga_n clusters ($n= 1-4$) from gas phase Gallium.

	ΔE_{0K} (kJ/mol _{Ga})
$Ga(g) \longrightarrow Ga_2$	-82
$Ga(g) \longrightarrow Ga_3$	-124
$Ga(g) \longrightarrow Ga_4$	-153
$Ga(g) \longrightarrow Ga_{bulk}$	-290

Table S2. Adsorption energy of Ga_n clusters on ZSM-5 ($Ga_n/ZSM-5$, 2 Al sites) and on pure-silica ZSM-5 ($Ga_n/ZSM-5$, no Al sites). DFT calculations performed using PBE-D3(BJ) at 0 K.

Ga_n	Adsorption energy, ΔE_{0K} (kJ/mol _{cluster})	
	$Ga_n/ZSM-5$ (2 Al sites) (R1)	$Ga_n/ZSM-5$ (no Al sites) (R2)
Ga_1	-116	-58
Ga_2	-127	-92
Ga_3	-142	-98

Ga ₄	-143	-105
-----------------	------	------

Comment 4. Line 80: "the application of liquid metal": I know the system is operating above the melting point of Gallium, but what proof is there that the Ga is actually in liquid state?

Reply 4: The transition of Ga to the liquid state can be many different techniques. For example, the presence of a negative peak at 33 °C over Ga/ZSM-5 sample in the heat flow of TG analysis indicates the endothermic melting transition of Ga in the sample.

Revision 4 (P12): "Ga is changing colour by interaction with zeolite, however, it is still in the form of metal covered by a thin oxide layer. The presence of Ga in the metal state can be confirmed by TG analysis, where it shows an endothermic peak during melting at 33 °C (Figure S4, SI)."

Figure S4. Heat flow during TGA analysis of Ga/ZSM-5 before and after reaction

Comment 5. Line 89 Figure 1. I believe the cartoon-description of the coke species inside the zeolite pores is misleading. The smallest aromatic molecule (i.e. coke molecule) is benzene, which is about 5 Å in size, which is almost the entire radius of the zeolite pore. Coke precursors in MTH are believed to get stuck inside the zeolite pores and thereby block the pore system. That does not change here. Additional coke deposits form outside the crystal and block the pore entrances. I do believe the authors demonstrate this latter coke may be reduced, but I do not believe the coke-precursor transport by the liquid Ga.

Reply 5: The additional study of Ga/ZSM-5 by the cross-sectional TEM analysis shows that metallic Ga can penetrate to distances of about 20 nm inside of the crystal. Thus, indeed, the effect of enhanced stability can be assigned to the continuous extraction of coke mainly from these acid sites, which can provide stable performance of the catalyst. We have performed an additional TEM study of a deactivated catalyst treated by Ga to confirm that treatment by Ga results in the migration of part of carbon to external zones of zeolite crystals to Ga. It is interesting to note that pure Ga nanoparticles also contain carbon on the surface indicating the high extraction ability of carbon by Ga nanoparticles.

Revision 5: (P13) *“STEM-HAADF and elemental mapping images of as-prepared Ga/ZSM-5 show the decoration of zeolite crystals by small-size Ga nanoparticles (Figure 3a-c). Cutting of the sample by cryo-ultramicrotome demonstrates penetration of Ga to the distance of only about 20 nm in zeolite crystal (Figure 3d-f, S9, SI), which could be explained by diffusion limitations for deeper penetration of Ga inside of the pores.”*

(P16) *“Microscopy analysis of deactivated ZSM-5 after Ga treatment shows a significant change in the distribution of carbon in comparison with the parent ZSM-5_20h sample (Figure 5c,f, Figure S19, SI). There is more carbon localized in the intercrystalline zone between zeolite crystals together with Ga. It is interesting to note that large Ga particles present in the sample after treatment also contain carbon on the surface (Figure S20, SI) indicating the high affinity of carbon to metallic Ga.*

The localization of carbon in the intercrystalline zone with Ga after the reaction could

be due to the extraction of coke from the zeolite, but this hypothesis is less likely due to the limited penetration capacity of Ga into the zeolite pores. Another explanation could be a stable catalytic performance mainly in this intercrystalline zone, which explains the coke generation. The role of Ga could involve continuous regeneration of acid sites in this area through the elimination of coke species until Ga and acid sites are completely saturated by carbon.”

Figure 5. Representative HAADF (a-c) and EDS elemental mapping of C (d-f) of ZSM-5_20h (a,d), Ga/ZSM-5_20h (b, e) and ZSM-5_20h+Ga (c, f).

Figure S9. STEM-HAADF images after cutting with cryo-ultramicrotome of Ga/ZSM-5 sample.

Figure S19. STEM-HAADF and EDS elemental mapping of Ga, C and superposition of Si and C in the sample ZSM-5_20h+Ga

Figure S20. *STEM-HAADF and EDS elemental mapping of Ga and C for Ga nanoparticle in the sample ZSM-5_20h+Ga*

Comment 6. Line 104: "The unsupported Ga particles were removed by sieving" could you provide some more detail?

Reply 6: We used a 250 μm stainless steel mesh sieve to remove the large Ga droplets not consumed by zeolite.

Revision 6: (P5) *"After the treatment, the unsupported Ga droplets were separated by sieving using a 250 μm stainless steel mesh sieve, the rest of the powder sample was collected and denoted as Ga/ZSM-5."*

Comment 7. Line 124: "the BET surface area was determined": Is BET a good method to probe the pore volume of microporous zeolites?

Reply 6: The BET model is often used to calculate the surface area of zeolite but it does not apply well to microporous materials. However, according to recent publications BET surface area is well representative of micropore surface areas of microporous materials (<https://doi.org/10.1021/acs.langmuir.8b02144>). In our case, we followed the variation of BET surface area after the reaction.

Comment 8. Line 147: reference 35. Something went wrong, there is no information in the reference list.

Reply 8: The reference has been corrected.

Comment 9. "In the case of the test with H₂, 18 mL/min of N₂ was mixed with 2 mL/min of H₂." : When exactly was this mixture used? Also during heating? Can the authors provide an estimate of the H₂/MeOH ratio?

Reply 9: The mixture of H₂ and N₂ has been formed before saturation by CH₃OH with subsequent addition to the reactor, where it was heated. The ratio of H₂/CH₃OH is about 2.

Revision 9 (P9): *“In the case of the test with H₂, 18 mL/min of N₂ was mixed with 2 mL/min of H₂ before entering the reactor with a ratio of H₂ to methanol of about 2. All gas lines after the reactor were heated to 100 °C, and the reaction product analysis was carried out using a Bruker GC456 GC equipped with a Rt-Q-BOND column (30 m, 0.32 mm ID, 10 μm) and an FID.”*

Comment 10. Line 200, figure 2A. There are interesting differences in the induction periods. Can you elaborate a bit more on that?

Reply 10: The differences in the induction period can be due to the differences in the initiation of the carbon pool mechanism for methanol activation over ZSM-5. Most probably in the presence of hydrogen, it takes more time to form methylated aromatic species to initiate the synthesis of hydrocarbons. It requires additional study.

Comment 11. "50 mg of catalyst or 50 mg of ZSM-5 mixed with 125 mg of liquid metal" This is not very clear. Does that imply there is always 50 gram of zeolite? Or should we read this another way?

Reply 11: Yes, it is always 50 mg of ZSM-5 in the reactor in pure form or mixed with liquid metals. The description has been corrected.

Revision 11 (P11): *“Reaction conditions: 400 °C, catalyst containing 50 mg of ZSM-5, 1.4 g methanol ·g⁻¹_{ZSM-5}·hour⁻¹.”*

Comment 12. Line 227: Why were the other metals not tested with H₂ in the feed?

Reply 12: We have additionally tested BiSn as the best alloy after Ga in the presence of H₂. This information has been added in SI.

Revision 12: *“The presence of hydrogen did not affect significantly the catalytic stability of BiSn+ZSM-5 in comparison with Ga+ZSM-5 (Figure S2, SI).”*

Figure S2. Products selectivity for ZSM-5 mixing with different liquid metals. Reaction conditions: 400 °C, catalyst containing 50 mg of ZSM-5, 1.4 g methanol · g⁻¹_{ZSM-5}·hour⁻¹.

Comment 13. Line 236: "It should be noted that Ga+ZSM-5 exhibited the highest Ga content following the reaction". Does this imply some of the liquid metals disappear?

Why, where? Boiling points are much higher than the process temperatures.

Reply 13: The chemical analysis has been performed for the zeolite part in the catalyst by separation of a metal part after reaction by sieving.

Revision 13: (P12) *“The chemical analysis of Ga/ZSM-5 and Ga+ZSM-5 separated from Ga particles by sieving shows that they contain 24 and 14.9 wt. % Ga, respectively.”*

Comment 14. Line 246: "penetration of Ga into the zeolite crystal": I do not believe the 2D TEM is convincing proof of penetration into the pores. The XRD peaks at low angle are almost unchanged, which would indicate there is no incorporation of any significant species into the pores.

Reply 14: Indeed, 2D TEM is not convincing to prove the presence of Ga inside of the pores of zeolite ZSM-5. We have tried to perform tomography TEM, however, it requires using long electron beam exposure for imaging Ga in zeolites, which results in radiation damage to zeolite structure. This is why to solve this problem we decided to perform HAADF imaging by cutting zeolite nanoparticles after freezing in liquid nitrogen with cryo-ultramicrotome. The images (**Figure 3, S9, SI**) show that Ga penetrates in zeolite crystals to a distance of about 20 nm. The penetration ability is higher in intercrystallite pores where zeolite crystals are in contact with each other. Thus, indeed, Ga cannot fill the pores of zeolite and the stable catalytic performance most probably is provided by the external acid site and pore mouths of zeolites.

Revision 14: *See revision to comment 5.*

Comment 15. Line 249: " the lower atomic radius" surely the atomic radius is not the only parameter that influences whether an atom in a liquid will enter pores. What about surface energy, contact angle, viscosity, etc.?

Reply 15: The comparison of Ga and In in terms of surface energy and viscosity shows that the surface energy of Ga and In have comparable values at 713 and 569 mJ/m² (10.1016/j.apsusc.2011.01.123), respectively. The viscosity of Ga is significantly lower in comparison with In and has values at 450 K of 1.016 and 1.748 mPa s (J. Phys. Chem. Ref. Data, Vol. 41, No. 3, 2012), respectively. It could also explain the high mobility on

the surface of zeolite and the fast decoration of zeolite crystals by Ga.

Revision 15: (P13) *“The lowest atomic radius of Ga (1.35 Å) in comparison with other liquid metals (In: 1.93 Å, Sn: 2.17 Å, Bi: 1.63 Å) and low viscosity of Ga (Ga: 1.016 mPa s, In: 1.748 mPa s) at 177 °C⁴⁵ can provide the highest access to the pores of ZSM-5 (5.4 Å × 5.6 Å).”*

Comment 16. Line 255: "undetectable at the TEM resolution". Earlier on the page you indicate some of the Ga goes missing, is that not a more logical explanation for not seeing it in TEM?

Reply 16: We mean here that Ga particles during in-situ TEM experiment disappear due to spreading in zeolite particles by diffusion on the surface of zeolite particles and inside of the pores. We have corrected the text to make it more clear.

Revision 16: *“During the activation with N₂ and exposure to the methanol, the initial Ga droplets deformed and redispersed to form smaller Ga particles or Ga-containing species on the surface and at the entrance of zeolite pores.”*

Comment 17. Line 262: why was the Ga/ZSM-5 not used?

Reply 17: TG analysis over Ga/ZSM-5_20h has been added in the Figures with text. For the sake of clarity, we have removed the TG analysis of the samples after 1h of the test.

Revision 17 (P15): *“TG analysis (Figure 2c, Figure S15, SI) performed on spent ZSM-5, Ga/ZSM-5 and Ga+ZSM-5 after 20 h on stream shows two stages of the weight loss (10 wt % in total) in ZSM-5_20h at about 100 and 550 °C, which can be ascribed according to the literature³⁷ to the desorption of water and burning of condensed the graphitic coke species, respectively. Ga/ZSM-5_20h and Ga+ZSM-5_20h also show the formation of graphitic coke, however, the weight loss is about ~4 times lower than that for ZSM-5_20h, suggesting Ga can suppress the formation of coke species over ZSM-5 and the catalyst deactivation.”*

Figure 2. Catalytic conversion (a) and selectivity to the products after 8 h of reaction (b) of ZSM-5, Ga₂O₃/ZSM-5, Ga+ZSM-5 and Ga/ZSM-5 in the methanol-to-hydrocarbon reaction. Reaction conditions: 400 °C, catalyst containing 50 mg of ZSM-5, 1.4 g methanol · g⁻¹ZSM-5·hour⁻¹. TGA analysis (c) and low-temperature N₂ adsorption (d) before and after 20 h of the catalytic tests for ZSM-5, Ga+ZSM-5 and Ga/ZSM-5.

Comment 18. Line 272: Same, why was the Ga/ZSM-5 not used for this comparison?

Reply 18: N₂ adsorption analysis over Ga/ZSM-5 has been added in the Figures with text.

Revision 17: “The ZSM-5 after N₂ activation at 450 °C for 1 h possesses a surface area of 339 m²/g, while the Ga+ZSM-5 and Ga/ZSM-5 samples show a decrease of the surface area to 266 and 160 m²/g, respectively, due to the introduction of Ga species

into the pores of zeolite and dilution effect of zeolite with metallic gallium. Coke deposition over ZSM-5 after 20 h resulted in a significant decrease in the surface area to 62 m²/g. In contrast, the surface area is almost retained on Ga+ZSM-5_20h and Ga/ZSM-5_20h.”

Comment 19. Line 280: This is not the experiment I would have used to make any statements about regeneration. Adding Ga may help remove the coke outside the crystals, but not the coke inside the crystals.

Reply 19: Indeed, this experiment has been used only to confirm the effect of Ga on the extraction of coke species from zeolite. Indeed, it seems that coke is mostly extracted from the surface and sub-surface layer, where most Ga is localized

Revision 19: See revision to comment 1.

Comment 20. Line 290: "slowing down coke deposition in the micropores". That would be enough by itself, the "washing out" is not necessary to explain the better performance (and in my opinion, not a logical explanation).

Reply 20: Yes, we agree that we do not have enough statements about the removal of coke from inside of the pores considering that Ga does not penetrate deep enough into the crystal. It seems more logical to assume that the reaction proceeds in this area and Ga prevents deactivation of the acid sites by removal of coke.

Revision 20: (P17) *“The localization of carbon in the intercrystalline zone with Ga after the reaction could be due to the extraction of coke from the zeolite, but this hypothesis is less likely due to the limited penetration capacity of Ga into the zeolite pores. Another explanation could be a stable catalytic performance mainly in this intercrystalline zone, which explains the coke generation. The role of Ga could involve continuous regeneration of acid sites in this area through the elimination of coke species until Ga and acid sites are completely saturated by carbon.”*

Comment 21. Line 332. The formation of hydrogen indicates removal of BAS sites. Does the addition of hydrogen during the experiment reinstate them?

Reply 21: The in-situ test in FTIR cell by reduction of Ga/ZSM-5 by hydrogen at 450 °C with subsequent Py adsorption demonstrates the same quantity of acid sites indicating the absence of reduction of cationic Ga back to metallic state.

Revision 21 (P 21): “It is interesting to note that cationic Ga is most probably not affected by hydrogen due to the same amount of adsorbed Py after pretreatment of the Ga/ZSM-5 catalyst in H₂ at 450 °C (Figure 6c,d).”

Figure 6. In situ toluene desorption DRIFT spectra over ZSM-5 and Ga/ZSM-5 catalysts at room temperature for 0 and 12 min (a). The residual percentage of toluene according to the peak-intensity changes (signal at ~1496 cm⁻¹) over ZSM-5 and Ga/ZSM-5 catalysts for 0–12 min (b). FTIR spectra of activated and spent ZSM-5 and Ga/ZSM-5 catalysts with and without H₂ treatment: region of hydroxyl groups vibrations (c) and region of pyridine adsorption vibrations (d).

Comment 22. Line 343 The TPR-temperature are measured in a dynamic system, and can not be directly compared to the reaction conditions.

Reply 22: TPR has been used here to compare the reducibility of the catalysts. The reduction of Ga in Ga/ZSM-5 has been additionally confirmed by XPS and TGA analysis.

Comment 23. Line 356: "It should result in the pushing out of carbon species and the release of acid sites of ZSM-5." I do not believe this. Ga outside the crystal may help removing carbon extra-crystalline carbon species. The only results on acid sites you show indicate a drop in acid sites.

Reply 23: We agree that the main contribution of Ga in the extraction of coke is on the external surface of zeolite and at the entrances of the pores. The small number of available acid sites refreshed by metallic Ga seems to play a crucial role in providing stable catalytic performance of the catalyst.

Revision 23: (P21) *"However, according to TEM and FTIR results, metallic Ga localized over the external surface of zeolite and at the entrance of the pores effectively promotes more stable methanol conversion in MTH process by slowing deposition and facilitating the desorption of carbon species. The internal acid sites should contribute less to the catalytic performance due to blockage of the pores by liquid Ga and their deactivation by reaction with Ga. At the same time, Ga at the external surface of zeolite provides continuous refreshing of the acid sites in this zone by removing coke species and liberating the acid sites."*

Comment 24. Line 359: "The dynamic nature of Ga promotes further migration of carbon from the inside of the pores to the external surface of zeolite crystals" This would imply there is a driving force for Ga to move to the outside of the crystal.

Reply 24: See Reply 23.

Comment 25. Line 367: I do not believe you have provided convincing evidence that the Ga is inside the zeolite pores.

Reply 25: See reply 5.

Comment 26. Line 269: This is a very nice result, but we would need to see the effect of oxidative regeneration.

Reply 26: See reply 1.

Comment 27. Line 370: "facial" do you mean "facile"?

Reply 27: It has been corrected.

Comment 28. Line 463: Information reference 35 is missing

Reply 28: It has been added.

Supporting information

Comment 29. In general: supporting text explaining what it is we see or are supposed to be seeing would be helpful.

Reply 29: The explanation has been added to some of the Figures of SI.

Comment 30. Figure S1: consider using the same x-axis for all plots. Why does ZSM-5H₂ shows such a long induction period? Where are the results for Ga/ZSM-5 and Ga+ZSM-5 with H₂?

Reply 30: Ga/ZSM-5 and Ga+ZSM-5 have been added in SI. One of the reasons for the long induction period of ZSM-5(H₂) can be the slow formation of carbon pool in ZSM-5 zeolite in the presence of hydrogen.

Figure S1. Conversions and selectivities versus time on stream for Ga/ZSM-5, Ga+ZSM-5, H-ZSM-5 catalysts with and without hydrogen. Reaction conditions: 400 °C, catalyst containing 50 mg of ZSM-5, 1.4 g methanol · g⁻¹_{catalyst} or ZSM-5 · hour⁻¹

Comment 31. Figure S4: I believe you mention you have indications for Ga-deposits

on the reactor walls. Where do we see that?

Reply 31: We can see the Ga mirror on the surface of the reactor after the reaction.

Comment 32. Figure S5: Please explain the observations in the SI text. Where does all the other metal go?

Reply 32: Other metals have not been incorporated in the zeolite phase and can be removed by sieving without impact on the composition of the zeolite phase.

Comment 33. Figure S6. The ratio of intensities for the peaks below 10 degrees 2 theta to the other peaks does not seem to change. That is a clear indication the Ga is not in the pores.

Reply 33: According to TEM analysis we confirm that Ga is mainly at the external surface and penetrates only about 20 nm inside the zeolite.

Comment 34. Figure S8. Please explain what we are supposed to look for.

Reply 34: The explaining text has been added in SI and Figure was corrected for clarity.

Revision 34: *“The test was carried out by analyzing the different zones of the catalyst at room temperature in a flow of N₂ at room temperature, 450 °C and at 400 °C in methanol. The images show a change in the morphology of the Ga droplets due to interaction with the zeolite. This takes the form of a gradual decrease in their intensity, sometimes resulting in their disappearance (red shapes). This phenomenon seems to depend on droplet size, as large particles do not appear to have suffered such a loss throughout the experiment.”*

Figure S11: Evolution of the Ga/ZSM-5 sample during the in situ STEM observation under N_2 at r.t., 450 °C and under methanol/ N_2 mixture at 400 °C

Comment 35. Figure S9. I see no difference between these three images. 3D TEM or tilt series would be more convincing.

Reply 35: The explaining text has been added in SI and Figure was corrected for clarity.

Comment 36. Figure S10. Why is Ga/ZSM-5 not shown?

Reply 36: It has been added.

Figure S15. Weight derivatives for thermogravimetric analysis of ZSM-5, Ga/ZSM-5 and Ga+ZSM-5 after MTH on stream for 20 h.

Comment 37. Figure S11: Is the amount zeolite in both experiments the same? Or is the zeolite diluted in the Ga+used ZSM-5 case?

Reply 37: Yes, the amount of zeolite is the same in both experiments.

Comment 38. Figure S16: I would consider showing only the relevant parts of the GC-spectra. The blue text (?) is unreadable, and may be irrelevant. Consider re-designing this graph.

Reply 38: The Figures have been redesigned to show only relevant information.

Figure S23. GC spectra of the hydrogen formation by treating ZSM-5 with Ga. 100 mg ZSM-5 mixing with 100 mg of gallium, 10 bar N_2 , 250 °C.

Comment 39. Figure S17: Why does it take so long before we see hydrogen. Why do we see hydrogen with neat ZSM-5? Can it be something else? Do you have GCMS to prove it is H_2 ? or just GC retention times?

Reply 39: There is a peak of hydrogen which has been confirmed using the standard gas mixture. The delay in appearance could be explained by the slow diffusion of Ga inside of the pores for interaction with acid sites of zeolite.

Comment 40. Figure S18: (e): why does the metallic Ga go down with H₂-addition?
(g): where is the second O 2s peak?

Reply 40: The deconvolution colours of O peaks have been corrected to provide two peaks of oxygen. The decrease in Ga peak can be attributed to the decrease of the dispersion of Ga after the reduction of Ga₂O₃ in the catalyst.

Figure S25. XPS of Ga 3d+O 2s of Ga₂O₃/ZSM-5, Ga+ZSM-5 and Ga/ZSM-5 before and after reaction.

Reviewer #2

The manuscript titled "Liquid Metals for Boosting Stability of Zeolite Catalysts in the Conversion of Methanol to Hydrocarbons" addresses an intriguing topic in catalysis. The authors explore the use of liquid metals to enhance the stability of zeolite catalysts during the conversion of methanol to hydrocarbons, liquid metal seems to have become the armor of ZSM-5 zeolites, preventing surface coking, and even has a certain reduction effect for the already coked zeolites, such effect was assigned to the high affinity of Ga to carbon, which is quite compelling. This study presents a novel approach by incorporating liquid metals into the catalytic system, and this concept has the potential to significantly improve the stability of zeolite catalysts, which is a critical challenge currently. The use of liquid metals as promoters offers new avenues for enhancing the catalytic performance and extending the catalyst lifetime. This innovation opens up possibilities for developing more efficient and robust catalytic systems for methanol conversion. However, there are still some queries which should be clarified:

Comment 1. In page 6 line 103-109, the only difference between the preparation of Ga/ZSM-5 and Ga+ZSM-5 appears to be that Ga+ZSM-5 is prepared without heating and sieving procedure. Could you explain the purpose of heating? Some research had reported that metal oxides seem soluble in gallium under specific condition. Does external heat promote the fusion of ZSM-5(SiO_2 or Al_2O_3) with gallium?

Reply 1: The mixing of Ga with ZSM-5 by preliminary heating in a batch reactor under stirring should provide a more uniform distribution of Ga through zeolite in comparison with mechanical mixing in a fixed bed reactor. Indeed, according to BET analysis (**Figure 2**), the interaction is significantly stronger in the case of Ga/ZSM-5 catalyst. It resulted in a more stable catalytic performance of Ga/ZSM-5 in the presence of hydrogen. The fusion of zeolite with liquid metal has not been observed according to XRD and TEM analysis, however, we could detect the distribution of Ga on the surface

of zeolite with partial diffusion in the pores of the zeolite (**Figure 3**). The fusion takes place usually in the case of the reduction of oxide by Ga to form an alloy, however, silicon oxide seems difficult to reduce under these conditions.

Revision 1 (P6): *“Ga/ZSM-5 was obtained by mixing the 50 mg of H-ZSM-5 with metal Ga in a weight ratio of 1:1 then stirring and heating the mixture at 250 °C under 10 bar of N₂ overnight. The heating was used to facilitate gallium migration over the zeolite.”*

Comment 2. Fig S3 shows that the Ga/ZSM-5 catalyst completely lost the metallic luster of gallium, but the powder looked moist and varied in particle size. I wonder if there is a grinding or sieving process before filling it in the quartz tube.

Reply 2: Indeed, Ga is changing colour by interaction with zeolite, however, it is still in the form of metal covered by a thin oxide layer. The presence of Ga in the metal state can be confirmed by TG analysis, where it shows an endothermic peak during melting. There is an excess of Ga which has not been consumed by zeolite. The excess of Ga has been removed by sieving.

Revision 2 (P12): *“Ga is changing colour by interaction with zeolite, however, it is still in the form of metal covered by a thin oxide layer. The presence of Ga in the metal state can be confirmed by TG analysis, where it shows an endothermic peak during melting at 33 °C (Figure S4, SI).”*

(P6) *“After the treatment, the unsupported Ga droplets were separated by sieving using a 250 μm stainless steel mesh sieve, the rest of the powder sample was collected and denoted as Ga/ZSM-5.”*

Figure S4. Heat flow during TGA analysis of Ga/ZSM-5 before and after reaction

Comment 3. In page 18 line 334-343, what is the proximate factor in the deactivation of ZSM-5 due to gallium oxidation?

Reply 3: The catalyst deactivation principally results from carbon deposition. It is very hard to evaluate the effect of Ga oxidation on the deactivation of the catalyst because the samples before and after contact with Ga contain different amounts of acid sites. It seems the stable activity in the presence of Ga is provided by a relatively small amount of acid sites close to the external surface which are continuously “washed” by liquid metal Ga. The saturation of this zone by coke results finally in full deactivation of the catalyst.

Revision 3 (P 20): “The catalyst deactivation principally results from carbon deposition. These results, however, support our assumption that the deactivation of Ga-modified ZSM-5 could be also induced by the oxidation of Ga by water during the catalytic reaction. It should suppress the regeneration effect of metallic Ga on the acid sites.”

- [1] J. Mitra, X. Zhou, T. Rauchfuss, Pd/C-catalyzed reactions of HMF: decarbonylation, hydrogenation, and hydrogenolysis, *Green Chemistry*, 17 (2015) 307-313.
- [2] T. Thananattachon, T.B. Rauchfuss, Efficient Production of the Liquid Fuel 2,5-Dimethylfuran from Fructose Using Formic Acid as a Reagent, *Angewandte Chemie International Edition*, 49 (2010) 6616-6618.
- [3] A.B. Gawade, M.S. Tiwari, G.D. Yadav, Biobased Green Process: Selective Hydrogenation of 5-Hydroxymethylfurfural to 2,5-Dimethyl Furan under Mild Conditions Using Pd-Cs_{2.5}H_{0.5}PW₁₂O₄₀/K-10 Clay, *ACS Sustainable Chemistry & Engineering*, 4 (2016) 4113-4123.
- [4] J.J. Wiesfeld, M. Kim, K. Nakajima, E.J.M. Hensen, Selective hydrogenation of 5-hydroxymethylfurfural and its acetal with 1,3-propanediol to 2,5-bis(hydroxymethyl)furan using supported rhenium-promoted nickel catalysts in water, *Green Chemistry*, 22 (2020) 1229-1238.
- [5] B.M. Matsagar, H.-L. Sung, J.-Y. Yeh, C.-T. Chen, K.C.W. Wu, One-step hydrogenolysis of 5-hydroxymethylfurfural to 1,2,6-hexanetriol using a Pt@MIL-53-derived Pt@Al₂O₃ catalyst and NaBH₄ in aqueous media, *Sustainable Energy & Fuels*, 5 (2021) 4087-4094.
- [6] V.A. Tur, A.V. Okotrub, Y.V. Shubin, B.V. Senkovskiy, L.G. Bulusheva, Chlorination of perforated graphite via interaction with thionylchloride, *physica status solidi (b)*, 251 (2014) 2613-2619.

REVIEWERS' COMMENTS

Reviewer #2 (Remarks to the Author):

This manuscript is highly commendable for its innovative approach in using gallium as a promoter to enhance the catalytic stability of methanol conversion to hydrocarbons. The study addresses the common issue of rapid deactivation in catalysts such as ZSM-5 and SAPO-34, which often suffer from the formation of bulk coke compounds and require regular regeneration treatments. By physically mixing ZSM-5 zeolite with liquid gallium, the researchers were able to significantly prolong the lifetime of the methanol conversion reaction, with an increase in longevity of up to 14 times. This breakthrough provides a new strategy for designing and preparing zeolite catalysts with improved resistance to deactivation. The findings also shed light on the role of gallium species, both cationic and oxide, as Lewis acid sites that facilitate the dehydrogenation-aromatization process. Overall, this research contributes to the advancement of catalytic stability in methanol conversion and opens up new possibilities for the modification of non-metallic catalysts using liquid metals.

The revised manuscript had properly explained my queries and explained the revision given, I would grant this manuscript acceptance.

Reviewer #3 (Remarks to the Author):

The manuscript titled "Enhancing Stability of ZSM-5 Zeolite Catalysts in Methanol-to-Hydrocarbons Conversion through Liquid Metal Addition" by Zhou et al. explores the impact of liquid metals on the stability of ZSM-5 catalysts in the process of methanol-to-hydrocarbons (MTH) conversion. The authors employ an unconventional approach to modify the base catalyst by directly incorporating metallic Ga, In, Bi/In, and Bi/Sn through diffusion into the pre-prepared ZSM-5 zeolite. Catalysts prepared in this manner exhibit a significantly increased lifetime, up to approximately 14 times. The authors attribute this enhanced stability to the Ga-induced promotion of aromatic species desorption, consequently slowing coke deposition.

Following the first revision, the authors made substantial enhancements to the manuscript by:

- (1) Providing additional physico-chemical characterization to substantiate the diffusion of gallium within the zeolite;
- (2) Offering a DFT-based elucidation for the penetration of Ga into the zeolite;
- (3) Rectifying minor discrepancies that impede the clarity of the text.

The manuscript is ready for publication in Nature Communications; nevertheless, I believe that incorporating some minor clarifications will notably enhance the scientific robustness of the paper.

Comment 1.

It would be advantageous to include a brief reference to the mechanism of the MTH conversion and use it to articulate the impact of Ga more distinctly. Differentiate the influence of Ga at both the macro- and atomic levels and draw comparisons between your hypothesis and the selectivity of Ga-induced

catalysts. For instance, elucidate the reasons behind Ga/ZSM-5 exhibiting a higher proportion of aromatics and C5+ in comparison to Ga+ZSM-5.

Comment 2

It would be a good idea to consolidate tables S1 and S2, introducing a new column for the sum of the formation and adsorption energies. Additionally, juxtapose these energies with the formation energy of bulk Ga. This way you provide a clear semi-quantitative DFT-based explanation for the thermodynamic favorability of the diffusion process. It would also be beneficial to compare the formation energy of bulk Ga with the experimental sublimation enthalpy, acknowledging potential significant disparities.

Comment 3

It may not be advisable to compare the Ga/ZSM acid center and Ga/HCl interaction energies in this paper without an explanation for the observed effect which is required for the deeper understanding of the Ga/ZSM system, and this aspect appears somewhat incongruous. It is recommended to omit this comparison from the main text.

Comment 4

Figure 2d presents challenges in readability. First, it's hard to single out a pair of the experiments corresponding to the same catalyst. I recommend employing color differentiation for catalysts and using various plotting symbols to indicate the time of catalytic tests. Additionally, consider rephrasing "before 20h of the catalytic test" to something akin to "the initial catalyst" for improved clarity.

REVIEWERS' COMMENTS

We thank the reviewers for the comments to our work. We made point-by-point response to these comments and revised the manuscript accordingly. All major changes have been highlighted in the manuscript and SI.

Reviewer #2 (Remarks to the Author):

This manuscript is highly commendable for its innovative approach in using gallium as a promoter to enhance the catalytic stability of methanol conversion to hydrocarbons. The study addresses the common issue of rapid deactivation in catalysts such as ZSM-5 and SAPO-34, which often suffer from the formation of bulk coke compounds and require regular regeneration treatments. By physically mixing ZSM-5 zeolite with liquid gallium, the researchers were able to significantly prolong the lifetime of the methanol conversion reaction, with an increase in longevity of up to 14 times. This breakthrough provides a new strategy for designing and preparing zeolite catalysts with improved resistance to deactivation. The findings also shed light on the role of gallium species, both cationic and oxide, as Lewis acid sites that facilitate the dehydrogenation-aromatization process. Overall, this research contributes to the advancement of catalytic stability in methanol conversion and opens up new possibilities for the modification of non-metallic catalysts using liquid metals.

The revised manuscript had properly explained my queries and explained the revision given, I would grant this manuscript acceptance.

Reply: We are very grateful to the reviewer for positive and pertinent comments, constructive suggestions and kind support. Following previous revisions, the manuscript has been considerably strengthened.

Reviewer #3 (Remarks to the Author):

The manuscript titled "Enhancing Stability of ZSM-5 Zeolite Catalysts in Methanol-to-Hydrocarbons Conversion through Liquid Metal Addition" by Zhou et al. explores the impact of liquid metals on the stability of ZSM-5 catalysts in the process of methanol-to-hydrocarbons (MTH) conversion. The authors employ an unconventional approach

to modify the base catalyst by directly incorporating metallic Ga, In, Bi/In, and Bi/Sn through diffusion into the pre-prepared ZSM-5 zeolite. Catalysts prepared in this manner exhibit a significantly increased lifetime, up to approximately 14 times. The authors attribute this enhanced stability to the Ga-induced promotion of aromatic species desorption, consequently slowing coke deposition.

Following the first revision, the authors made substantial enhancements to the manuscript by:

- (1) Providing additional physico-chemical characterization to substantiate the diffusion of gallium within the zeolite;
- (2) Offering a DFT-based elucidation for the penetration of Ga into the zeolite;
- (3) Rectifying minor discrepancies that impede the clarity of the text.

The manuscript is ready for publication in Nature Communications; nevertheless, I believe that incorporating some minor clarifications will notably enhance the scientific robustness of the paper.

Reply: We thank Reviewer 3 for the positive evaluation of our Manuscript and acknowledgment of its suitable content, insights, analysis, and organization. In what follows, we address the revision requests of Reviewer 3.

Comment 1. It would be advantageous to include a brief reference to the mechanism of the MTH conversion and use it to articulate the impact of Ga more distinctly. Differentiate the influence of Ga at both the macro- and atomic levels and draw comparisons between your hypothesis and the selectivity of Ga-induced catalysts. For instance, elucidate the reasons behind Ga/ZSM-5 exhibiting a higher proportion of aromatics and C5+ in comparison to Ga+ZSM-5.

Reply 1: We have provided more information concerning the mechanism of MTH and role of Ga in the reaction.

Revision 1: P3 *“According to the latest results, the MTH reaction proceeds according to a double-cycle mechanism with the production of paraffins and olefins by the olefin cycle and ethylene and aromatics by the aromatics cycle.”* *“Framework Ga in zeolite also proved favorable to aromatization reactions²⁶.”*

P18 *“However, these species affect the selectivity of the reaction in N₂ atmosphere, resulting in an increased yield of aromatics for Ga₂O₃/ZSM-5 and Ga/ZSM-5 compared to H-ZSM-5, which could be the result of the dehydrogenation activity on Ga oxide (Figure 2). Ga+ZSM-5 provides lower selectivity to aromatics most probably due to less uniform distribution of Ga in the sample and lower content of oxidized Ga in this case. Interestingly, the hydrogenation activity on Ga/ZSM-5 and Ga+ZSM-5 is considerably enhanced in the presence of hydrogen with higher paraffins production compared to ZSM-5 and Ga₂O₃/ZSM-5 (Figure 2). This could indicate hydrogenation behavior of cationic Ga in ZSM-5 or oxidized Ga on the Ga metal surface.”*

Comment 2. It would be a good idea to consolidate tables S1 and S2, introducing a new column for the sum of the formation and adsorption energies. Additionally, juxtapose these energies with the formation energy of bulk Ga. This way you provide a clear semi-quantitative DFT-based explanation for the thermodynamic favorability of the diffusion process. It would also be beneficial to compare the formation energy of bulk Ga with the experimental sublimation enthalpy, acknowledging potential significant disparities.

Reply 2: Table S1, SI and Table S2, SI from the supporting information were recombined and a new column was also included as suggested. The formation energy of gas phase Ga_n clusters (n= 1-4) and the adsorption energy of Ga_n clusters on ZSM-5 (Ga_n/ZSM-5, 2 Al sites) were used to calculate the formation energy of Ga_n/ZSM-5 (new column). The adsorption of Ga_n clusters in Silicalite-1 (pure Si form, no Al sites) was also added to Table S1, SI in order to make the comparison to Ga_n/ZSM-5. Relative to the experimental value of sublimation enthalpy, it is present at the explanation found in the supporting information in the following form: *“The stability of these clusters increases with the number of Ga atoms, converging to the sublimation energy for bulk Gallium, -290 kJ/mol_{Ga}. The latter DFT PBE-D3(BJ) value corresponds reasonably well with the experimental standard latent heat of sublimation, 277 kJ/mol.⁷ This high value is also consistent with the high temperature required to vaporize Ga.”*; and it is also included below Table S1, SI.

Table S1. Formation energy of gas phase Ga_n clusters, adsorption energy of Ga_n clusters on ZSM-5 ($Ga_n/ZSM-5$, 2 Al sites) and on Silicalite-1 ($Ga_n/Si-1$, no Al sites), and the formation energy of adsorbed Ga_n clusters on ZSM-5. DFT calculations performed using PBE-D3(BJ) at 0 K.

	$Ga_n(g)$	Ga_n cluster	Ga_n cluster	$Ga_n/ZSM-5$
Ga_n	formation energy (a) (kJ/mol Ga)	adsorption energy on ZSM-5 (b) (kJ/mol $_{cluster}$)	adsorption energy on Si-1 (c) (kJ/mol $_{cluster}$)	formation energy (a)+(b/n) (kJ/mol Ga)
Ga_1	0	-116	-58	-116
Ga_2	-82	-127	-92	-146
Ga_3	-124	-142	-98	-171
Ga_4	-153	-143	-105	-189
Ga_{Bulk}	-290*			

* $\Delta H_{sublimation}(Ga) = 277$ kJ/mol Ga (experimental value at 1 atm).

Comment 3. It may not be advisable to compare the Ga/ZSM acid center and Ga/HCl interaction energies in this paper without an explanation for the observed effect which is required for the deeper understanding of the Ga/ZSM system, and this aspect appears somewhat incongruous. It is recommended to omit this comparison from the main text.

Reply: Thanks for the comments. We initially made the comparison of the interaction energies between Ga/ZSM-5 and Ga/HCl to distinguish the nature of acid sites, however, strong interaction was not observed for Ga with HCl (-41 kJ/mol vs. -116 to -143 kJ/mol). We agree that omitting this not very relevant comparison enhances the clarity.

Comment 4. Figure 2d presents challenges in readability. First, it's hard to single out a pair of the experiments corresponding to the same catalyst. I recommend employing color differentiation for catalysts and using various plotting symbols to indicate the time of catalytic tests. Additionally, consider rephrasing "before 20h of the catalytic test" to something akin to "the initial catalyst" for improved clarity.

Reply: We have redesigned the Figure 2 to make clearer the attribution of the catalysts. The phrase "before and after 20h of the catalytic test" has been changed to the "initial and after 20 h on stream".